# Decoding m⁶Am by simultaneous transcription-start mapping and methylation quantification

**Jianheng Fox Liu, Ben R Hawley[†], Luke S Nicholson, Samie R Jaffrey***

Department of Pharmacology, Weill Cornell Medicine, Cornell University, New York, United States

## eLife Assessment

This study presents a new quantitative method, CROWN-seq, to map the cap-adjacent RNA modification N6,2'-O-dimethyladenosine (m6Am) with single nucleotide resolution. Using thoughtful controls and well-validated reagents, the authors provide **compelling** evidence that the method is reliable and reproducible. Additionally, the study provides **important** evidence that m6Am may increase transcription in modified mRNAs. However, the data only demonstrates a correlation between m6Am and transcriptional regulation rather than causality. Overall, this study is poised to advance m6Am research, being of broad interest to the RNA biology and gene regulation fields.

**\*For correspondence:**
srj2003@med.cornell.edu

**Present address:** †Engage Bio, San Carlos, United States

**Abstract** $N^6$,2'-*O*-dimethyladenosine (m⁶Am) is a modified nucleotide located at the first transcribed position in mRNA and snRNA that is essential for diverse physiological processes. m⁶Am mapping methods assume each gene uses a single start nucleotide. However, gene transcription usually involves multiple start sites, generating numerous 5' isoforms. Thus, gene-level annotations cannot capture the diversity of m⁶Am modification in the transcriptome. Here, we describe CROWN-seq, which simultaneously identifies transcription-start nucleotides and quantifies m⁶Am stoichiometry for each 5' isoform that initiates with adenosine. Using CROWN-seq, we map the m⁶Am landscape in nine human cell lines. Our findings reveal that m⁶Am is nearly always a high stoichiometry modification, with only a small subset of cellular mRNAs showing lower m⁶Am stoichiometry. We find that m⁶Am is associated with increased transcript expression and provide evidence that m⁶Am may be linked to transcription initiation associated with specific promoter sequences and initiation mechanisms. These data suggest a potential new function for m⁶Am in influencing transcription.

## Introduction

m⁶Am ($N^6$,2'-*O*-dimethyladenosine) is the most common modified nucleotide in mRNA. m⁶Am is found specifically at the first transcribed position of mRNAs, termed the transcription-start nucleotide (TSN), which reflects the transcription-start site (TSS) in DNA. During transcription, the TSN typically acquires a 2'-*O*-methyl modification (*Wang et al., 2019*; *Galloway et al., 2020*), which is deposited by CMTR1 (Cap-specific mRNA nucleoside-2'-*O*-methyltransferase 1; *Bélanger et al., 2010*). In the case of mRNAs that use adenosine as the TSN, the initial 2'-*O*-methyladenosine (Am) can be further methylated on the $N^6$ position of the adenine nucleobase to form m⁶Am by PCIF1 (phosphorylated CTD interacting factor 1; *Akichika et al., 2019*).

Studies using PCIF1 depletion (i.e. m⁶Am depletion) have revealed that m⁶Am has important roles in cell physiology. In normal cells, PCIF1 depletion does not appear to affect cell growth or viability

(*Akichika et al., 2019*). However, in oxidative stress conditions, PCIF1 deficiency leads to impaired cell growth (*Akichika et al., 2019*). In cancer cells, PCIF1 depletion markedly enhances cell death during anti-PD1 therapy (*Wang et al., 2023b*). During viral infection, PCIF1 depletion results in increased HIV replication (*Zhang et al., 2021b*), impaired SARS-Cov-2 infection (*Wang et al., 2023a*), and increased VSV immunogenicity (*Tartell et al., 2021*). These studies indicate that m⁶Am has important roles in diverse cellular contexts.

A major goal has been to identify and characterize the m⁶Am- and Am-containing mRNAs. Initial chromatographic studies in the 1970s demonstrated that cellular mRNAs can exist in m⁶Am and Am forms, with the Am form being more predominant (*Wei et al., 1975*). To map m⁶Am modified genes, several antibody-based methods were developed, including miCLIP (*Mauer et al., 2019*), m⁶Am-seq (*Sun et al., 2021*), m6ACE-seq (*Koh et al., 2019*), and m6Am-exo-seq (*Sendinc et al., 2019*). These methods can identify m⁶Am sites (miCLIP and m6ACE-seq), m⁶Am peaks (m⁶Am-seq), or m⁶Am containing genes (m⁶Am-exo-seq). Am genes were identified when m⁶Am was not detected but the reported TSN in publicly available datasets was A. The remaining genes were annotated as Gm, Cm, or Um based on public TSS annotations.

However, despite these transcriptome-wide m⁶Am maps, the effect of m⁶Am on mRNA is unclear. By examining m⁶Am genes, along with the change in mRNA stability and translation, small and inconsistent effects have been observed from different labs (*Akichika et al., 2019*; *Sendinc et al., 2019*; *Boulias et al., 2019*; *Mauer et al., 2017*).

We considered the possibility that the difficulty in establishing m⁶Am function may be due to flaws in the way that genes are designated as m⁶Am genes. Previous m⁶Am mapping studies treated m⁶Am like other modified nucleotides, which are internal. In studies of internal nucleotides, such as m⁶A, isoform diversity is generally not considered since these different isoforms rarely impact the detection of the nucleotide. In contrast, m⁶Am is highly affected by isoform diversity since it is located at the 5' end. Most genes generate multiple transcript 5' isoforms that each use a different TSS (*Noguchi et al., 2017*). However, previous m⁶Am mapping studies assumed that each gene has a single TSN, whose identity was based on existing gene annotations. Therefore, existing m⁶Am maps that assign a specific start nucleotide to each gene cannot be accurate, since most genes produce a range of transcripts, with possibly multiple start nucleotides. For this reason, m⁶Am mapping and functional studies of m⁶Am need to be performed in a way that considers the 5' isoform diversity of most genes.

Another concern is that the existing m⁶Am mapping studies designated each gene as either m⁶Am or Am. However, it is possible that m⁶Am levels can be variable, with only a fraction of transcripts containing m⁶Am and the remainder containing Am. Stoichiometric maps of m⁶Am can potentially reveal the degree to which a transcript would be influenced by m⁶Am-dependent pathways.

Quantitative m⁶Am mapping is especially important for small nuclear RNAs (snRNAs), which also contain m⁶Am at their TSN. Initial biochemical characterization of snRNAs revealed that the first nucleotide was Am (*Dönmez et al., 2004*), but subsequent studies showed that nearly half of all snRNAs are initially methylated to m⁶Am, and then demethylated by FTO to Am (*Mauer et al., 2019*; *Koh et al., 2019*). Thus, m⁶Am is a transient intermediate in snRNA biogenesis. Notably, m⁶Am levels can be highly regulated in snRNAs (*Mauer et al., 2019*), however, current m⁶Am mapping methods are unable to quantify changes in m⁶Am stoichiometry.

To understand the transcriptome-wide distribution of m⁶Am, we developed CROWN-seq (**C**onversion **R**esistance detection **O**n **W**hole-transcriptomic transcription-start **N**⁶,2'-*O*-dimethyladenosine by **seq**uencing), an antibody-free quantitative m⁶Am mapping method. Using CROWN-seq, we define the overall repertoire of 5' isoforms for each gene, and the specific isoforms that use m⁶Am as the TSN across nine different cell lines. We find that annotations of genes based on a single start nucleotide do not capture the diversity of 5' transcript isoforms for most genes. Instead, m⁶Am is more accurately assessed for each 5' isoform separately. Nearly all A-initiated transcript isoforms have very high m⁶Am stoichiometry, and that transcripts containing Am as the TSN are relatively rare. Transcript isoforms that contain m⁶Am are more highly expressed, and loss of m⁶Am due to depletion of PCIF1 leads to reduced expression of transcript isoforms containing m⁶Am. However, we find that this effect is not due to decreased mRNA stability. Instead, the depletion of PCIF1 affects transcripts based on upstream core promoter elements. Our data suggest that transcription mechanisms that utilize specific core promoter sequences achieve high expression, which might be linked to a transcription-promoting effect of m⁶Am. Overall, our quantitative transcriptome-wide transcription-start nucleotide

m⁶Am maps reveal a markedly distinct m⁶Am profile different from previously measured, show that m⁶Am is the predominant modified nucleotide relative to Am in mRNA, and suggest roles of m⁶Am in transcription.

## Results

### ReCappable-seq reveals high transcript isoform diversity at the 5' end

In previous m⁶Am mapping studies, genes were annotated to be m⁶Am, Am, Gm, Cm, or Um (*Akichika et al., 2019*; *Mauer et al., 2019*; *Sendinc et al., 2019*; *Boulias et al., 2019*), based on the assumption that each gene has one major transcription-start nucleotide. To determine how often genes are characterized by a single major TSS we used ReCappable-seq (*Yan et al., 2022*), a method for quantitative measurement of transcription-start sites. ReCappable-seq is similar to traditional TSS-seq methods which involve ligation of an oligonucleotide to the 5' end of mRNAs (*Yamashita et al., 2011*), thus precisely marking the TSN. However, ReCappable-seq adds an enrichment step for capped mRNA fragments to significantly reduce background signals from internal sites that are derived from RNA cleavage. Thus, ReCappable-seq provides a highly sensitive and precise mapping of TSNs at single-nucleotide resolution (see Materials and methods).

By analyzing ReCappable-seq data in HEK293T cells, we found that protein-coding genes tend to have multiple TSNs. Among the 9199 genes analyzed, we identified 87,624 TSNs (see Materials and methods). On average, a gene uses 9.5±9 (mean and s.d., hereafter) TSNs (*Figure 1A*). Only ~9% of genes contain a single TSN (*Figure 1A*). Thus, most genes cannot be characterized by a single start nucleotide.

As an example, *SRSF1*, which was previously classified as an m⁶Am gene, has ~37 different 5' isoforms in HEK293T cells, of which ~41.5% do not use an A-TSN (*Figure 1B*). As another example, *ADAR* was previously classified as a Gm gene, but ~71.0% of transcripts use A-TSNs (*Figure 1C*). These observations are not artifacts of ReCappable-seq because similar results were also found using other TSN mapping methods (*Figure 1—figure supplement 1A, B*).

Conceivably m⁶Am genes produce multiple 5' isoforms, but the isoforms predominantly use A-TSNs. If this were the case, then the gene could indeed be considered an m⁶Am gene if all the A-TSNs were methylated to m⁶Am. We considered a gene to be predominantly composed of A-TSNs if >80% of transcripts start with A. Using this criterion, we found that only ~24% of m⁶Am genes determined by miCLIP (*Boulias et al., 2019*) are primarily composed of A-TSNs (*Figure 1D*). Similar observations were also found in other m⁶Am mapping methods (*Akichika et al., 2019*; *Sendinc et al., 2019*, *Figure 1—figure supplement 1C, D*).

Our ReCappable-seq analysis also suggested that previous m⁶Am mapping methods may not have detected the diversity of m⁶Am in the transcriptome. ReCappable-seq identified many more A-TSNs than the total number of previously mapped m⁶Am sites. For example, in both *SRSF1* and *ADAR*, many A-TSNs are seen using ReCappable-seq, however, m⁶Am signals were only found at a few of these A-TSNs by either miCLIP (*Boulias et al., 2019*), m⁶Am-seq (*Sun et al., 2021*), or m6ACE-seq (*Koh et al., 2019*, *Figure 1B and C*). This might suggest that only a few A-TSNs are m⁶Am modified. However, it is also possible that the antibody-based mapping methods do not have the resolution or sensitivity to distinguish between m⁶Am at different 5' isoforms. Notably, previous m⁶Am mapping studies exhibited very low overlap with each other. miCLIP (*Boulias et al., 2019*), m⁶Am-seq (*Sun et al., 2021*), and m6ACE-seq (*Koh et al., 2019*) together identified 7480 m⁶Am sites in HEK293T cells (*Figure 1—figure supplement 1E*). Among these sites, only 1.1% (84) are found in all three methods and 9.7% (728) are found in at least two studies (*Figure 1—figure supplement 1E*). Taken together, these data demonstrate a variety of concerns about existing m⁶Am mapping studies.

### CROWN-seq integrates TSN mapping and m⁶Am quantification

To understand the distribution of m⁶Am in the transcriptome, we sought to develop a method to identify the entire repertoire of TSNs among all the 5' transcript isoforms for each gene. In this way, we can identify the specific 5' isoforms for each gene that contain m⁶Am. Additionally, we wanted a quantitative method rather than the qualitative assessment provided by previous antibody-based methods. Recently, chemical methods using sodium nitrite were developed for m⁶A analysis (*Liu et al., 2023*; *Mahdavi-Amiri et al., 2021*; *Werner et al., 2021*). This method identifies m⁶A by chemically

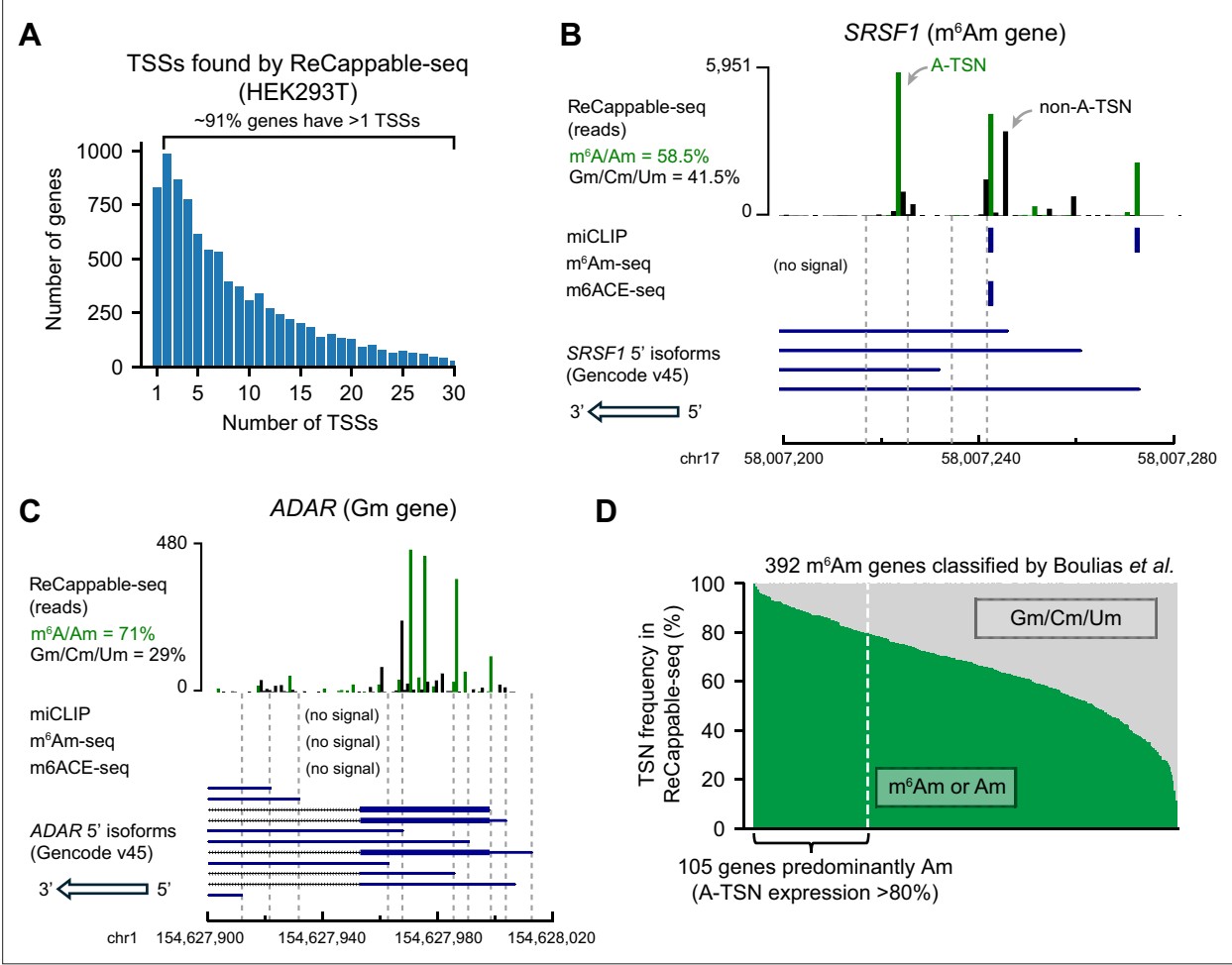

**Figure 1.** Many m⁶Am genes are mistakenly annotated. (**A**) Genes tend to have multiple TSSs. Shown is a histogram displaying the number of TSSs per protein-coding gene in HEK293T cells. TSSs (87,624 TSSs from 9199 genes) were mapped using ReCappable-seq. These TSSs have expression levels ≥1 TPM (transcription-start nucleotide per million). (**B, C**) Examples of genes that were mistakenly classified by previous studies (**_Boulias et al., 2019_**). *SRSF1* (**B**) was previously designated as m⁶Am because of the miCLIP signals overlapping with A-TSSs. However, based on ReCappable-seq, ~41.5% of the reads are mapped to non-A-TSSs in *SRSF1*. Notably, one of the most expressed m⁶Am A-TSS (chr17:58,007,228) was mistakenly considered as internal m⁶A because this position was not previously annotated as a TSS (**_Linder et al., 2015_**). *ADAR* (**C**) was previously classified as Gm. There is no m⁶Am signal based on miCLIP (**_Boulias et al., 2019_**), m⁶Am-seq (**_Sun et al., 2021_**), or m6ACE-seq (**_Koh et al., 2019_**) mapped to *ADAR*. However, ~71% of the transcripts from *ADAR* are A-initiated. (**D**) Previously classified m⁶Am genes express considerable levels of non-A-initiated transcripts. Each column represents a gene previously classified as m⁶Am gene by miCLIP (**_Boulias et al., 2019_**). For each gene, the percentage of transcript isoforms starting with m⁶Am/Am (in green) or Gm/Cm/Um (in gray) are shown. The percentage was calculated by weighting each transcript isoform by its expression level. The TSN frequencies were obtained using ReCappable-seq in HEK293T cells.

The online version of this article includes the following figure supplement(s) for figure 1:

**Figure supplement 1.** Corresponding to *Figure 1*.

**Figure supplement 2.** Corresponding to *Figure 1*.

deaminating ('converting') unmethylated A's into inosines (I's), while leaving m⁶A's intact. During sequencing, the A-to-I conversions are readily detected because I's are reverse transcribed into G's. This approach leads to precise and robust m⁶A quantification (**_Liu et al., 2023_**). Because of the chemical similarity between m⁶Am and Am, we explored the potential use of sodium nitrite conversion to map and quantify m⁶Am.

We first asked if Am is susceptible to deamination by sodium nitrite. To test this, we performed sodium nitrite conversion on a m⁷G-ppp-Am-initiated transcript (see Materials and methods). We applied the sodium nitrite conversion protocol used in GLORI, which includes glyoxal treatment to prevent modification of guanosine residues (**_Liu et al., 2023_**). After sodium nitrite treatment, the RNA

was reverse transcribed and sequenced. The conversion rate of Am was quantified by counting A or G reads at the first nucleotide position. In this assay, Am was completely converted (*Figure 2—figure supplement 1A*), indicating that sodium nitrite efficiently converts Am and thus can be used for m⁶Am quantification.

We considered the possibility that GLORI data (*Liu et al., 2023*) could be mined to measure m⁶Am stoichiometry at previously mapped m⁶Am sites (*Boulias et al., 2019*). We noticed that many A's at these TSNs were highly converted in GLORI (*Figure 2—figure supplement 1B*), suggesting prevalent Am. This is inconsistent with mass spectrometry analysis of mRNA cap structures from us (*Wang et al., 2019*) and others (*Galloway et al., 2020*; *Akichika et al., 2019*), which has suggested that m⁶Am is very prevalent while Am is relatively rare in mRNA. A potential cause of the high level of Am at TSNs predicted by GLORI could be the extensive RNA fragmentation that occurs with sodium nitrite treatment. RNA fragments that have 5' ends that align to the TSNs of overlapping transcript isoforms can confound the measurement of m⁶Am stoichiometry (*Figure 2—figure supplement 1C, D*). Thus, GLORI cannot distinguish between true TSNs and internal bases that are found at the 5' end of RNA fragments. To overcome this limitation, we sought to develop a method that selectively analyzes A-TSNs and thereby removes the confounding effect of overlapping transcripts.

We developed CROWN-seq, which selectively analyzes TSNs throughout the transcriptome (*Figure 2A*). In this method, Am residues in mRNA are converted to Im using sodium nitrite. Next, we specifically isolate the 5' ends of mRNA by replacing the m⁷G cap with a desthiobiotin affinity tag using a decapping-and-recapping strategy (*Yan et al., 2022*). By enriching the m⁷G-proximal sequence, we can simultaneously sequence the TSN of all transcripts, including both m⁶Am and non-m⁶Am TSNs. This is conceptually different from existing m⁶Am mapping methods which only examine the m⁶Am transcripts. For A-TSNs, m⁶Am stoichiometry can be quantified by counting the number of A reads (reflecting m⁶Am) or G reads (reflecting Am). In this way, we not only obtain TSN locations but also m⁶Am stoichiometry in the same RNA molecule.

To increase the accuracy of m⁶Am quantification, we made several optimizations to the ReCappable-seq protocol to markedly increase the read depth of TSNs. These include steps for on-bead adapter ligation and the introduction of unique molecular identifiers (UMIs) in the library preparation (see Materials and methods).

## Benchmarking CROWN-seq using m⁶Am-modified standards

To test TSN enrichment in CROWN-seq, we used a m⁷G-ppp-m⁶Am standard spiked into cellular mRNA. Among three technical replicates, we observed that ~93% of the reads mapped to the TSN (*Figure 2B*), confirming the enrichment of TSN. To further assess the enrichment of TSNs, we performed GLORI on the same sample. However, in GLORI only a few reads map to the TSN (*Figure 2—figure supplement 1C*). Thus, the decapping-and-recapping approach markedly enriches for TSNs.

We next wanted to determine the quantitative accuracy of CROWN-seq. To test this, we performed CROWN-seq on a mixture of RNA standards with predefined ratios of m⁶Am and Am (see Materials and methods). We found a highly linear correlation between the expected m⁶Am levels and the observed non-conversion rates measured by CROWN-seq across three technical replicates (Pearson's $r$=0.992, *Figure 2C*). Taken together, CROWN-seq achieves both precise TSS mapping and m⁶Am quantification in m⁶Am standards.

## CROWN-seq markedly expands the number of mapped m⁶Am sites

To assess the ability of CROWN-seq to map and quantify m⁶Am throughout the transcriptome, we performed CROWN-seq on poly(A)-selected RNA from HEK293T. In total, we identified 219,195 high-confidence TSNs, of which 92,278 were A-TSNs (see Materials and methods). These TSNs were highly reproducible across biological and technical replicates (*Figure 2—figure supplement 1E*). Among the A-TSNs, 89,898 were from protein-coding genes, and 219 were from snRNA or snoRNA. Notably, among the mRNA A-TSNs, nearly all had high non-conversion rates (*Figure 2—figure supplement 1F*), indicating that nearly all A-TSNs contain high stoichiometry m⁶Am.

In contrast to previous m⁶Am mapping methods, CROWN-Seq reveals the diversity of TSNs among all the transcript isoforms for each gene. For example, in the case of *SRSF1*, m⁶Am is readily visible along with multiple other TSNs comprising Gm, Cm, or Um (*Figure 2D*). CROWN-seq also shows that A-TSNs can have intermediate m⁶Am stoichiometry. For example, *JUN* expresses a 5' transcript

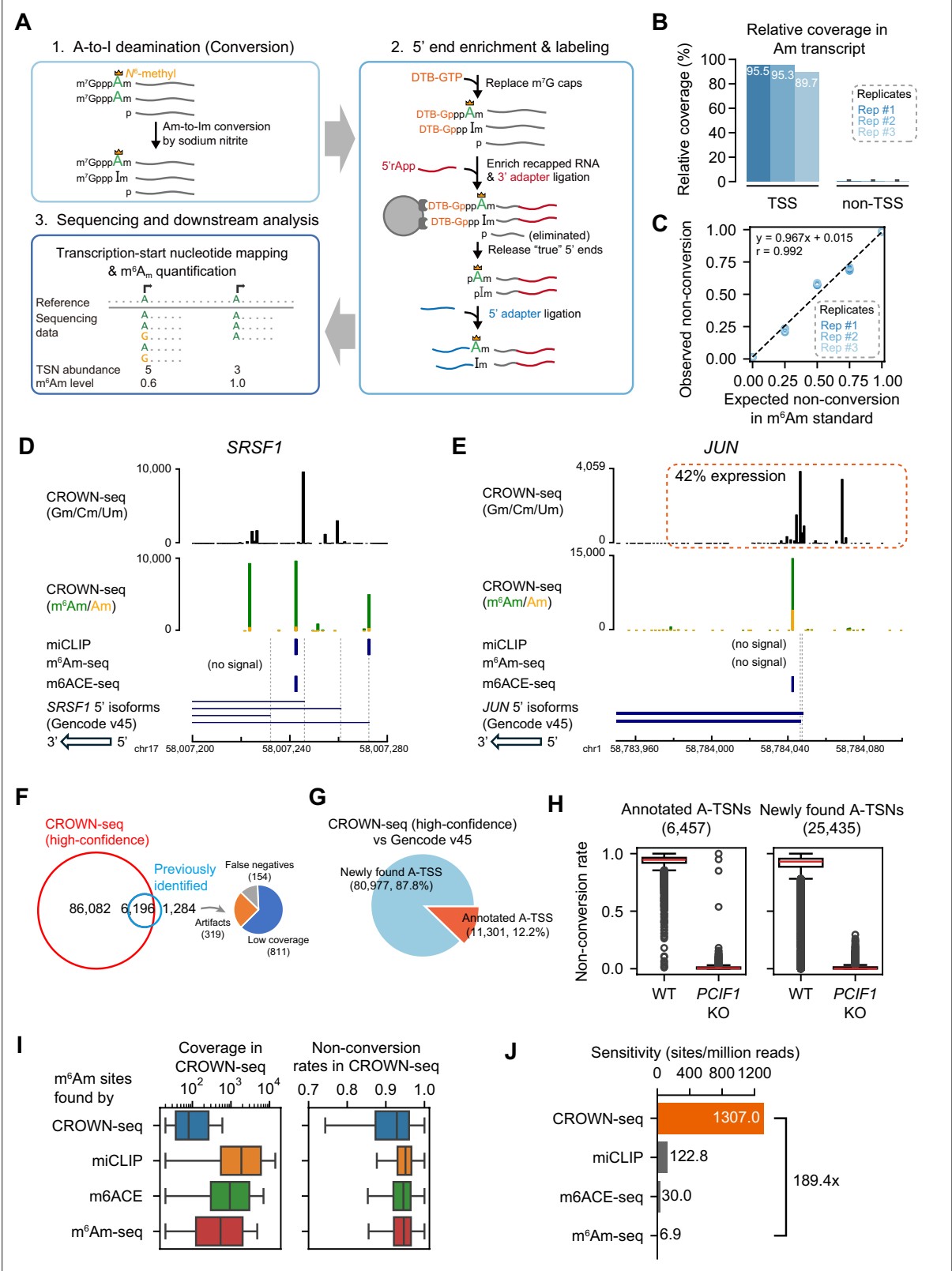

**Figure 2.** CROWN-seq correctly maps and quantifies m⁶Am. (**A**) Schematic of CROWN-seq. RNAs are firstly treated with sodium nitrite, which causes Am at the transcription-start position to be converted to Im. To isolate the TSN, m⁷G caps are replaced with 3'-desthiobiotin (DTB) caps. These DTB caps are enriched on streptavidin beads, while uncapped background RNA fragments are uncapped and washed away. After washing, an enriched pool of transcript 5' ends is released from the beads by cleaving the triphosphate bridge, leaving 5' monophosphate ends that are ligated to an adapter.

*Figure 2 continued on next page*

*Figure 2 continued*

After adapter ligation, cDNA was synthesized and amplified for Illumina sequencing. During sequencing, the converted sequences were aligned to a reference genome. The TSNs can be determined as the first base immediately after the 5' adapter sequence. To quantify m⁶Am stoichiometry, we count the number of A (m⁶Am) and G (Am) bases at the TSN position. (**B**) CROWN-seq enriches reads that contain the TSN. The relative coverage of reads mapped to the TSS and non-TSS regions across the m⁷G-ppp-Am-initiated RNA standard was calculated. The average relative coverage of reads that map to the TSS and to non-TSS positions are shown for three technical replicates. The 95% CI of the relative coverages is shown using error bars. (**C**) CROWN-seq exhibits high quantitative accuracy for measuring m⁶Am stoichiometry. RNA standards were prepared with 0%, 25%, 50%, 75%, and 100% m⁶Am stoichiometry. To make m⁶Am standards in different m⁶Am levels, we generated both Am transcripts and m⁶Am transcripts by in vitro transcription with cap analogs m⁷G-ppp-Am and m⁷G-ppp-m⁶Am. Five transcripts were made in the Am and m6Am form and mixed to achieve the indicated m⁶Am stoichiometry. These transcripts have identical 5' ends and different barcodes. Linear least-squares regression was performed to calculate the correlation between expected non-conversion rates and the observed average non-conversion rates for each standard. All TSNs shown in this plot have high sequencing coverage, ranging from 656 to 21,545 reads. (**D**) CROWN-seq results for *SRSF1*. CROWN-seq shows that 54.0% of *SRSF1* transcripts initiate with A. Among the A-initiated transcripts, 93.4% were resistant to conversion (A's, shown in green), and therefore m⁶Am. As a result, *SRSF1* has 50.4% m⁶Am transcripts, 3.6% Am transcripts, and 46.0% non-A-initiated transcripts. Notably, a previous miCLIP study identified an internal m⁶A site (***Linder et al., 2015***) which we found was m⁶Am at the TSN based on CROWN-seq. (**E**) CROWN-seq results for *JUN*. CROWN-seq shows that ~58% of *JUN* transcripts initiate with A. Unlike *SRSF1* which A-TSNs are highly methylated, *JUN* A-TSNs are only ~75% methylated. As a result, *JUN* has 43.5% m⁶Am transcripts, 14.5% Am transcripts, and 42% non-A-initiated transcripts. (**F**) CROWN-seq identifies most m⁶Am sites identified in previous studies. 7480 m⁶Am sites in HEK293T cells found either by miCLIP (***Boulias et al., 2019***), m⁶Am-seq (***Sun et al., 2021***), or m6ACE-seq (***Koh et al., 2019***) were analyzed. The high-confidence sites in CROWN-seq were defined as A-TSN with ≥20 unique mapped reads. The results shown are from HEK293T cells, which is the same cell line used in all previous studies. Among the 1,284 sites uniquely found in other studies, 811 sites are also mapped by CROWN-seq but at lower coverage (1–19 reads); 319 sites are mapped very far (>100 nt) away from any TSS annotation and thus can be considered as false positives; the remaining 154 sites are mapped very closely to known TSSs and may be false negative results in CROWN-seq. (**G**) Many A-TSNs identified in CROWN-seq in HEK293T cells are not annotated. In this analysis, A-TSSs in (**F**) were intersected with the TSS annotation in Gencode v45. Only 12.2% of A-TSSs found by CROWN-seq are previously annotated. (**H**) CROWN-seq exhibits high accuracy in TSN discovery. In this analysis, we compared the non-conversion of A-TSNs between wild-type and *PCIF1* knockout cells. For the 6,457 A-TSNs annotated by Gencode v45, most of them have high non-conversion rates in wild-type cells and very low non-conversion rates in *PCIF1* knockout cells, indicating correct TSN mapping. Similar to the annotated TSNs, 25,435 newly found A-TSNs were also found to have differential m⁶Am between wild-type and *PCIF1* knockout. Thus, these newly found A-TSNs were also mostly true positives. In this analysis, only A-TSNs mapped by at least 20 reads in both wild-type and *PCIF1* knockout HEK293T cells were used. (**I**) The previously identified m⁶Am sites are biasedly in higher expression and higher m⁶Am stoichiometry. Shown are the sequencing coverage (left) and non-conversion rates (right) of different sets of m⁶Am sites in HEK293T CROWN-seq data. In total, 98,147 sites found by CROWN-seq, 2129 sites found by miCLIP (***Boulias et al., 2019***), 3693 sites found by m6ACE-seq (***Koh et al., 2019***), and 1610 sites found by m⁶Am-seq (***Sun et al., 2021***) are shown. (**J**) CROWN-seq has much higher sensitivity in m⁶Am discovery than all existing m⁶Am mapping methods. In this analysis, sensitivity is defined as m⁶Am/A-TSN found per million mapped reads. For CROWN-seq, sensitivity was defined as the slope of linear regression result between sequencing depth and A-TSN number among different samples in this study (see ***Figure 2—figure supplement 1G***). For other methods, sensitivity was defined as the number of reported m⁶Am sites over the number of reads in all libraries required for m⁶Am identification.

The online version of this article includes the following source data and figure supplement(s) for figure 2:

**Source data 1.** A comparison of m⁶Am mapping methods.

**Source data 2.** The design of m⁶Am standards.

**Figure supplement 1.** Corresponding to *Figure 2*.

isoform with an A-TSN, of which ~75% of transcript copies are m⁶Am modified (***Figure 2E***). Overall, CROWN-seq provides a comprehensive assessment of all TSNs in a gene and reveals the fraction of each A-TSN that is m⁶Am.

To confirm the accuracy of the mapped m⁶Am TSNs, we examined the 7480 m⁶Am sites reported by miCLIP (***Boulias et al., 2019***), m⁶Am-seq (***Sun et al., 2021***), or m6ACE-seq (***Koh et al., 2019***). Among these sites, the vast majority (~82.8%, 6196 of 7480) were also found among the high-confidence A-TSNs in CROWN-seq (***Figure 2F***). For the remaining 1284 sites, 811 are also found in CROWN-seq, but in lower sequencing depth; 319 were located far away (>100 nt) from any known TSSs and thus may be artifacts. Thus, CROWN-seq is highly reliable in detecting known m⁶Am sites. The low consistency between previous m⁶Am mapping studies likely reflects incomplete m⁶Am mapping in previous methods.

CROWN-seq clearly identified vastly more A-TSNs than all the other m⁶Am mapping methods combined (12.3-fold, 92,278 vs 7480). Notably, only ~12.2% of the newly found A-TSNs in CROWN-seq are annotated in Gencode v45 (***Figure 2G***, ***Figure 2—figure supplement 1G***), which primarily relies on CAGE data (see gene annotation guidelines by HAVANA ***Havana, 2025***). Notably, the newly identified TSNs with high coverage tend to overlap with or be located proximally to known TSSs, while the ones with low coverage tend to be located further to the known TSSs (***Figure 2—figure supplement***

*1G*). The newly identified A-TSNs could be artifacts or could be actual TSNs that were undetected by previous TSS-mapping studies. We suspect that these are true A-TSNs for two reasons: First, as part of the mapping criteria, a minimum of 20 independent reads across all replicates was required for TSN identification. Second, if these sites were RNA fragments, they would not contain m⁶Am. However, these A-TSNs show high stoichiometry of m⁶Am (i.e. non-conversion) in CROWN-seq (*Figure 2H*, *Figure 2—figure supplement 1F*) which is lost in *PCIF1* knockout cells (*Figure 2H*), except for some outliers such as TSNs of *S100A6*, *IFI27*, and *ALDH1A1*. Thus, the marked increase in the number of m⁶Am sites revealed by CROWN-seq reflects the preferential enrichment for mRNA 5′ ends, which leads to high sensitivity and read depth at TSNs transcriptome-wide.

In contrast to m⁶Am sites identified in CROWN-seq, m⁶Am that were identified in previous m⁶Am mapping methods tended to derive from high abundance transcripts or high abundance TSNs (*Figure 2I*). Because of the high read depth at TSNs, CROWN-seq enables the detection of m⁶Am at more m⁶Am sites than previous methods (*Figure 2J*, *Figure 2—figure supplement 1H*). Although we used a 20-read cutoff for mapping m⁶Am, m⁶Am sites identified with fewer reads are also likely to represent true TSNs. These m⁶Am TSNs typically show high non-conversion (e.g. 2 or 3 reads among a total of 3 reads) in HEK293T cells but zero non-conversions in *PCIF1* knockout cells (*Figure 2—figure supplement 1I*). The PCIF1 dependence of these sites is consistent with a true m⁶Am TSN and further highlights the sensitivity of CROWN-seq for mapping m⁶Am at TSNs.

## CROWN-seq reveals consistently high m⁶Am stoichiometry in mRNA across diverse human cell lines

Although our data showed that m⁶Am in mRNA generally exhibits very high stoichiometry (*Figure 2—figure supplement 1F*), we considered the possibility that these results were unique to HEK293T cells. Several studies have shown that PCIF1 expression can vary considerably in different cell lines (*Wang et al., 2023b*; *Li et al., 2023*), which may indicate that m⁶Am stoichiometry is dependent on the cell line. We therefore wanted to determine the m⁶Am landscape across cell lines with varying levels of PCIF1.

We selected several cell lines for this analysis. First, we chose HEK293T, A549, HepG2, and K562 cells, which have also been characterized using multiple orthogonal datasets (*Djebali et al., 2012*). Second, we selected colorectal cancer cells (i.e. HT-29 and HCT-116), since PCIF1 depletion in these cells affects their migration and sensitivity to immunotherapy (*Wang et al., 2023b*). These colorectal cancer cells have high PCIF1 expression based on western blotting and RT-qPCR, while the non-cancerous colon cell line CCD841 CoN has very low PCIF1 expression (*Wang et al., 2023b*). Third, we selected cells with very low CTBP2 expression, a proposed coactivator of PCIF1 (*Li et al., 2023*). These cells, which include Jurkat E6.1 and Huh-7, as well as the previously mentioned K562 and HepG2 cells, would be expected to have low m⁶Am levels based on their low CTBP2 expression (*Li et al., 2023*; *Figure 3—figure supplement 1A*).

For each cell line, we performed CROWN-seq using two to four biological/technical replicates. In total, we obtained 514 million aligned reads (*Supplementary file 1*). In each cell line, 14,650–58,768 mRNA A-TSNs with at least 50 reads were analyzed (*Supplementary file 1*). The 50-read threshold provides highly consistent quantification of m⁶Am stoichiometry between replicates (*Figure 3—figure supplement 1B*).

Quantification of m⁶Am across all TSNs showed that mRNA m⁶Am stoichiometry is generally high. For most of the cells, the average m⁶Am stoichiometry is 0.895±0.03 (*Figure 3A*), indicating high overall mRNA m⁶Am levels. Some cell lines, for example, Jurkat E6.1, HT-29, and Huh-7 cells show very high and less variable m⁶Am levels (0.933±0.1, 0.924±0.1, and 0.916±0.1, respectively); while other cell lines such as CCD841 CoN, HCT-116, and K-562 have relatively low and more variable m⁶Am levels (0.825±0.2, 0.877±0.1, and 0.891±0.1, respectively). It should be noted that in all cell lines, the m⁶Am stoichiometry is still high compared with other mRNA modifications (*Liu et al., 2023*).

We considered the possibility that the high m⁶Am stoichiometry might be caused by RNA structure that blocks access to sodium nitrite leading to non-conversion. However, essentially complete conversion was seen in *PCIF1* knockout HEK293T cells, which makes it likely that m⁶Am is the cause of non-conversions. Also, we found that A-TSNs completely converted in 5′ ends predicted to be highly structured, suggesting that RNA structure does not impair conversion in CROWN-seq (*Figure 3—figure supplement 1C*).

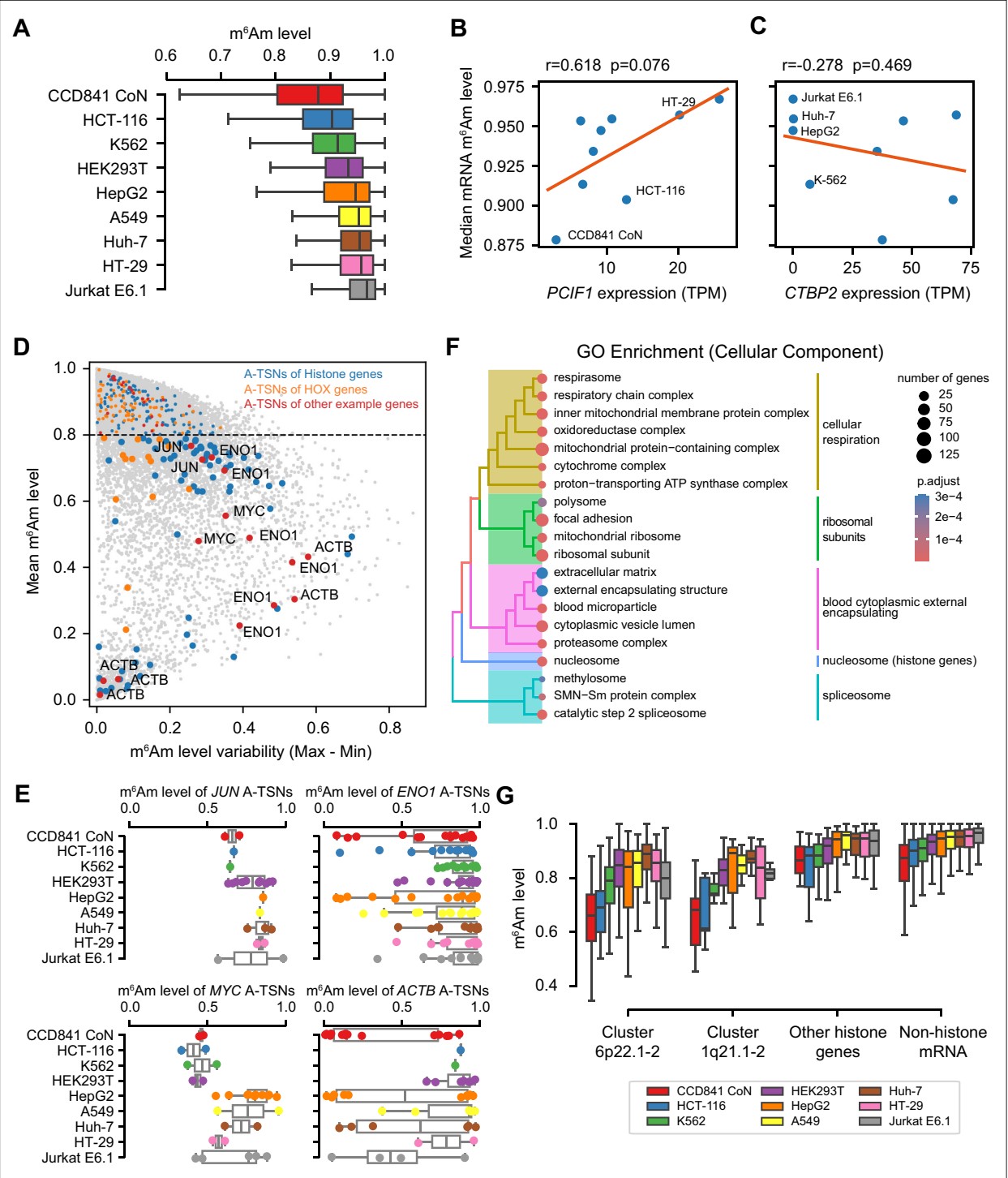

**Figure 3.** CROWN-seq reveals m⁶Am landscape in mRNA. (**A**) Boxplot showing the overall mRNA m⁶Am levels (i.e., m⁶Am stoichiometry) among different cell lines. Only m⁶Am sites with ≥50 reads mapped in at least one cell line were analyzed. (**B**) mRNA m⁶Am stoichiometry is positively correlated with PCIF1 expression. In this plot, *PCIF1* expression was estimated by the number of reads mapped to *PCIF1* TSSs. The read counts were normalized into transcription-start nucleotide per million (TPM) for gene expression comparison. Three cell lines (CCD841 CoN, HCT-116, and HT-29) whose *PCIF1* expression was estimated by western blots and RT-qPCR by *Wang et al., 2023b* are highlighted. Pearson's r and p-value in this analysis were obtained by linear regression. (**C**) Overall mRNA m⁶Am stoichiometry is not correlated with CTBP2 expression. Similar to (**B**), *CTBP2* expression was estimated by CROWN-seq. Four cell lines with very low *CTBP2* expression are highlighted. (**D**) Some A-TSNs have relatively low and more variable m⁶Am stoichiometry among cell lines. In this plot, the variability of the m⁶Am stoichiometry of a site, which is defined as the maximum m⁶Am subtracted by the minimum m⁶Am stoichiometry among all cell lines is shown on the X-axis; the average m⁶Am level of a site among all cell lines is shown on the Y-axis.

*Figure 3 continued on next page*

*Figure 3 continued*

Several example genes are indicated in different colors. (**E**) Boxplots and dotplots showing the m⁶Am levels of different A-TSNs in *JUN, ENO1, MYC*, and *ACTB*. These genes contain A-TSNs with relatively low m⁶Am stoichiometry. In this plot, the exact m⁶Am levels of individual A-TSNs are shown in dots, while the median and IQR of the m⁶Am levels are shown in boxplots. Only m⁶Am sites with ≥50 reads mapped were analyzed. (**F**) Gene Ontology enrichment (Cellular Components) of genes containing lowly methylated m⁶Am sites. (**G**) A-TSNs in histone genes tend to have relatively low m⁶Am stoichiometry. In this plot, histone genes are categorized by their genomic localizations. Histone gene cluster 6p22.1–2 and 1q21.1–2 are the two major histone gene clusters. For histone gene cluster 6p22.1–2, 55–173 A-TSNs are shown in different cell lines; for histone gene cluster 1q21.1–2, 9–14 A-TSNs are shown; for other histone genes, 24–109 A-TSNs are shown.

The online version of this article includes the following source data and figure supplement(s) for figure 3:

**Source data 1.** mRNA m⁶Am stoichiometries in different cell lines.

**Source data 2.** Gene Ontology enrichment of genes with relatively low m⁶Am sites.

**Figure supplement 1.** Corresponding to *Figure 3*.

The differences in m⁶Am stoichiometry are related to PCIF1 expression (*Figure 3B*, *Figure 3— figure supplement 1D*). For example, CCD841 CoN cells, which have very low PCIF1 expression based on our measurements (*Figure 3B*, *Figure 3—figure supplement 1D*) and previous measurements (*Wang et al., 2023b*), exhibit the lowest median m⁶Am stoichiometry at ~0.878. However, even this stoichiometry is still relatively high. Thus, m⁶Am levels are affected by PCIF1 expression, but m⁶Am can be considered as a high stoichiometry modification across all tested cell lines. On the other hand, the proposed PCIF1 coactivator CTBP2, exhibited a weak correlation to mRNA m⁶Am (*Figure 3C*).

## Several mRNAs show low m⁶Am stoichiometry

Although most A-TSNs in mRNA exhibit high m⁶Am stoichiometry, some exhibit stoichiometry below 0.8, and even below 0.5 (*Figure 3A*). To identify A-TSNs with low m⁶Am, we examined each A-TSN and calculated its average stoichiometry across all cell lines (*Figure 3D*). For each A-TSN, we also assessed its variability by calculating the range of m⁶Am levels measured across cell lines (*Figure 3D*). This analysis demonstrates that a significant subset of A-TSNs have low stoichiometry, with some showing variability depending on the cell type. For example, *JUN* contains a lowly methylated A-TSN, as shown above in HEK293T cells (*Figure 2E*), and also exhibits low stoichiometry in many other cell lines (*Figure 3E*). In addition, *ENO1*, *MYC*, and *ACTB* also show low m⁶Am stoichiometry in some of their A-TSNs (*Figure 3E*).

We next used Gene Ontology (GO) analysis to determine if the low m⁶Am A-TSNs are associated with specific cellular functions. The GO analysis of Cellular Component categories showed a marked enrichment of genes linked to cellular respiration, ribosomal subunits, spliceosome, and nucleosome (which are mostly histone genes; *Figure 3F*). Similar results were found in the Biological Processes GO analysis (*Figure 3—source data 2*). In addition to these genes, we also noticed HOX genes contain lowly methylated A-TSNs (*Figure 3—figure supplement 1E*).

Among all different gene categories, histone genes exhibited the lowest overall m⁶Am stoichiometry (*Figure 3G*). Notably, histone genes have unique mechanisms of gene expression. Many histone genes are located in gene clusters (i.e. clusters 6p22.1–2 and 1q21.1–2) and transcribed in histone locus bodies (*Marzluff and Koreski, 2017*). These clustered histone genes tend to contain upstream TATA-box and downstream T-rich sequences (*Figure 3—figure supplement 1F*). In contrast, non-clustered histone genes tend to have high m⁶Am stoichiometry (*Figure 3G*) and show different promoter sequence contexts (*Figure 3—figure supplement 1F*). This data suggests that transcription mechanisms might be important for determining m⁶Am stoichiometry.

## m⁶Am stoichiometry is linked to the sequence of core promoter

The differential methylation in histone genes based on their transcription mechanisms raises the possibility that transcription initiation mechanisms might affect m⁶Am stoichiometry. Because m⁶Am is the first nucleotide in mRNA, its deposition may be highly influenced by early transcription events. Notably, PCIF1 binds to RNA polymerase II (*Fan et al., 2003*) and is enriched in promoter regions (*Sugita et al., 2021*), which may be important for methylation of the 5' end of mRNAs. We therefore

considered the possibility that different transcription mechanisms may be linked to different levels of m⁶Am.

As a first test, we examined whether nucleotide preferences upstream (which would reflect sequences involved in transcription initiation) or downstream of the A-TSN are linked to m⁶Am stoichiometry. We binned A-TSNs based on the m⁶Am stoichiometry and examined nucleotide preferences at each position. Using this approach, we found that the nucleotides upstream of the A-TSN were markedly different for A-TSNs with low vs. high m⁶Am stoichiometry (*Figure 4—figure supplement 1A*). For example, at positions –4 and –1, there was a clear positive correlation between the use of C and m⁶Am stoichiometry (*Figure 4A*). The correlation of these nucleotide positions that lie in the promoter region to m⁶Am stoichiometry suggests that transcriptional mechanisms might influence m⁶Am stoichiometry.

We also observed strong nucleotide preferences at positions downstream of the A-TSN. These include nucleotide preferences at +2 (*Figure 4A*). These could reflect sequence preferences for PCIF1; however, this position is also part of transcription-initiation motifs (see below), and thus the contribution of transcription mechanisms and direct sequence preferences of PCIF1 are difficult to deconvolve.

To more directly determine specific transcription mechanisms linked to m⁶Am, we examined how specific sequence motifs around A-TSNs correlate with m⁶Am stoichiometry. We found markedly different sequence motifs surrounding highly methylated A-TSNs compared to lowly methylated A-TSNs (*Figure 4B*). A-TSNs with the highest m⁶Am stoichiometry (top 5th-percentile, 0.991 average stoichiometry) are enriched in an SSCA₊₁GC (S=C/G) motif, which is similar but distinct from the well-known BBCA₊₁BW (B=C/G/T, W=A/T) transcription initiator motif (*Haberle and Stark, 2018*), largely because of the C at the +3 position (*Figure 4B*). In contrast, the A-TSNs with the lowest m⁶Am stoichiometry (bottom 5th-percentile, 0.435 average stoichiometry) were enriched in an unconventional VA₊₁RR (V=A/C/G, *R*=A/G) motif (*Figure 4B*).

We next classified each A-TSN based on whether they use the SSCA₊₁GC or VA₊₁RR motifs, or if they contain the conventional BBCA₊₁BW and BA₊₁ Inr-like motifs (*Figure 4—figure supplement 1B*). This plot shows that BBCA₊₁BW and BA₊₁ motifs exhibit intermediate m⁶Am stoichiometry (*Figure 4C*). Overall, these data indicate that m⁶Am stoichiometry is strongly related to the TSS motif in the core promoter, which implies that m⁶Am formation is linked to the transcription initiation process.

Because transcription initiation is also affected by other elements in the core promoter (*Haberle and Stark, 2018*), we also asked whether these transcription-related elements, such as TATA-box and transcription factor-binding sites, are associated with higher or lower m⁶Am stoichiometry. We first analyzed the relationship between m⁶Am and elements including the TATA-box, BREu, BREd, and DCE (*Haberle and Stark, 2018*). In this analysis, A-TSNs from promoters containing TATA-box exhibited lower m⁶Am stoichiometry, especially those of histone genes (*Figure 4D*). On the other hand, other elements, such as BREu and BREd, which are motifs for recruitment of TFIIB (*Haberle and Stark, 2018*), and DCE, which binds by TAF1 (*Haberle and Stark, 2018*), showed little correlation with m⁶Am stoichiometry (*Figure 4—figure supplement 1C*). Thus, the presence of the TATA box exhibited the strongest effect and predicted lower m⁶Am stoichiometry.

We next analyzed the relationship between m⁶Am and transcription factor-binding sites (TFBS). To test this, we screened A-TSNs for the presence of 401 transcription-factor binding sites and examined the relationship between the binding sites and m⁶Am stoichiometry (see Materials and methods). Several TFBSs, such as those for NANOG and FOXJ3, exhibited a slight negative correlation to m⁶Am (*Figure 4—figure supplement 1D*); while other TFBS, such as SP2 and KLF4, exhibited a slight positive correlation to m⁶Am (*Figure 4—figure supplement 1E*). Overall, no specific TFBS exhibited a strong effect on m⁶Am stoichiometry (*Figure 4—figure supplement 1F*).

Taken together, our data show a linkage between transcriptional mechanisms and m⁶Am stoichiometry.

## m⁶Am does not substantially influence mRNA stability or translation

Previous studies sought to determine the effect of m⁶Am on mRNA stability and translation based on gene-level annotations of the starting nucleotide (*Akichika et al., 2019*; *Boulias et al., 2019*; *Mauer et al., 2017*; *Zhang et al., 2019*). However, the gene level annotations do not take into account the potential for many transcription-start nucleotides (*Figure 1—figure supplement 2*). Rather than using a binary metric of m⁶Am or non-m⁶Am, we developed a metric that reports the fraction of all TSNs for

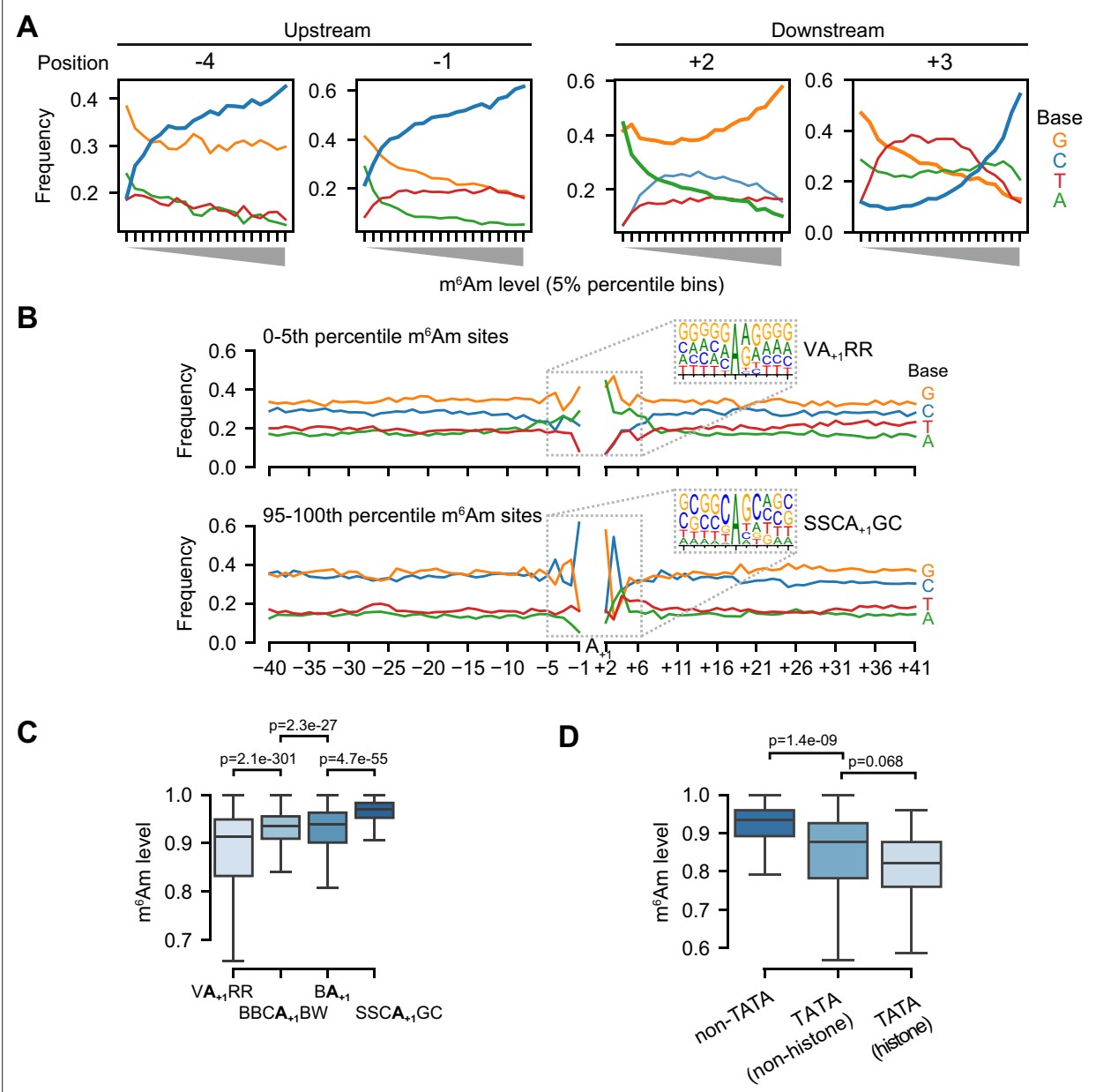

**Figure 4.** m⁶Am stoichiometry is related to core promoter sequence. (**A**) m⁶Am stoichiometry is related to base composition in both upstream and downstream of A-TSNs. In these plots, 58,723 A-TSNs are grouped into twenty 5-percentile bins (X-axis). For each bin, the frequency of A, T, C, and G bases at each position relative to the A-TSN are plotted on the Y-axis. Among different positions in the promoter region, C's in −4,−1, and +3, as well as G's in +2 are positively correlated with high m⁶Am; while A's in +2 are negatively correlated with high m⁶Am. Results for other promoter positions can be found in **Figure 4—figure supplement 1A**. (**B**) Motif analysis of A-TSNs with the lowest 5% m⁶Am stoichiometry (upper) and the A-TSNs with the highest 5% m⁶Am stoichiometry (lower). The core promoter region (–40 to +41) was screened for enriched motifs. The lowest 5% A-TSNs exhibited a VA₊₁RR TSS (V=A/C/G, R=A/G) motif, while the highest 5% A-TSNs exhibited a SSCA₊₁GC (S=C/G) motif. The sequence contexts for all A-TSNs are shown in **Figure 3—figure supplement 1F**. (**C**) A-TSNs expressed from different core promoters exhibit different m⁶Am stoichiometry. Core promoters containing the VA₊₁RR motif produce transcripts with relatively low m⁶Am stoichiometry. Transcripts using the SSCA₊₁GC motif exhibited relatively high m⁶Am stoichiometry. In comparison, the m⁶Am stoichiometry in conventional A-TSNs from either BBCA₊₁BW or BA₊₁ is also shown and exhibits intermediate m6Am stoichiometry. In this analysis, 14,788, 7981, 34,578, and 1376 A-TSNs were used for each of the four motifs. p-values, Student's t-test, two-sided. (**D**) TATA-box containing core promoters exhibit relatively low m⁶Am stoichiometry. For this analysis, the TATA-box is defined as TATAWAWR (**Haberle and Stark, 2018**). Because many TATA-boxes found in our A-TSN dataset are outside the classic −31 to −24 region, we extended the region for the TATA-box search to −36 to −19. Since histone genes preferentially contain TATA box, we separately plotted TATA-box-containing histone genes (N=155) and TATA-box-containing non-histone genes (N=28). 58,540 A-TSNs without TATA-box are also shown. p-values, Student's t-test, two-sided.

*Figure 4 continued on next page*

*Figure 4 continued*

The online version of this article includes the following figure supplement(s) for figure 4:

**Figure supplement 1.** Corresponding to *Figure 4*.

each gene that contains m⁶Am. This 'm⁶Am gene index' is the ratio of m⁶Am TSNs over all TSNs, as measured by CROWN-seq, for each gene. Using the m⁶Am gene index, we reanalyzed the previously published translation efficiency (*Akichika et al., 2019*; *Boulias et al., 2019*) and RNA stability (*Boulias et al., 2019*) data in HEK293T cells. We found that genes with low or high m⁶Am gene index do not show differences in translation (*Figure 5—figure supplement 1A, B*) or RNA stability (*Figure 5—figure supplement 1C*) in *PCIF1* knockout cells compared to wild-type.

## m⁶Am is involved in efficient transcription of A-initiated transcripts

We next wanted to examine other potential functions of m⁶Am. Although we found no clear effect of m⁶Am on mRNA stability, we asked if m⁶Am influences transcript expression levels. To test this, we quantified the abundance of each A-TSN isoform in HEK293T and A549 cells. For these experiments, we added a mixture of pre-capped ERCC spike-ins (see Materials and methods) to the RNA samples before performing TSN expression quantification by ReCappable-seq. This ERCC spike-in mixture calibrates sequencing results and increases TSN expression quantification accuracy (see Materials and methods).

In this analysis, we binned A-TSNs into percentiles based on their m⁶Am stoichiometry. Here, we could see that transcripts with the highest levels of m⁶Am also exhibited the highest overall expression levels (*Figure 5A*, *Figure 5—figure supplement 1D*). This suggests that m⁶Am is associated with higher transcript expression.

To determine if m⁶Am causes the increased expression of A-TSN transcripts, we measured the expression change for each A-TSN in wild-type and *PCIF1* knockout HEK293T and A549 cells. We found that A-TSNs with higher m⁶Am stoichiometry exhibit significantly reduced expression in *PCIF1* knockout, while A-TSNs with the lowest m⁶Am stoichiometry were almost unchanged (*Figure 5B*, *Figure 5—figure supplement 1F*). In contrast, G-TSNs were slightly increased in *PCIF1* knockout cells (*Figure 5C*, *Figure 5—figure supplement 1F*). These data suggest that m⁶Am promotes the expression of A-TSN transcripts.

We were surprised that PCIF1 depletion leads to a selective decrease in the expression of A-TSN transcripts in the highest percentile bin but had little to no effect in the other bins. Each bin has very high m⁶Am stoichiometry (~0.77 in the lowest bin and ~0.98 in the highest bin in HEK293T) (*Figure 5B*, *Figure 5—figure supplement 1E*). Thus, if m⁶Am is a stabilizing mark, we should see reduced expression in all bins. We therefore considered other possibilities that might explain why PCIF1 depletion affects transcript levels in some bins but not others.

An important difference between A-TSN in different bins is that they tend to use different TSS motifs (see *Figure 4C*). We therefore asked if the effect of m⁶Am depletion is linked to the TSS motifs. For this analysis, we classified A-TSNs based on the presence of SSCA$_{+1}$GC, VA$_{+1}$RR, or other TSS motifs (i.e. BBCA$_{+1}$GC and BA$_{+1}$). Here we found that the identity of the TSS motif was highly associated with the degree of transcript reduction in *PCIF1* knockout cells (*Figure 5D*, *Figure 5—figure supplement 1G*). Notably, transcripts that use the SSCA$_{+1}$GC motif showed the largest drop in expression. In contrast, A-TSNs that use the VA$_{+1}$RR TSS motif showed almost no change in expression in *PCIF1* knockout cells (*Figure 5D*, *Figure 5—figure supplement 1G*).

Taken together, these data suggest that the effect of m⁶Am is not related to mRNA stability but instead is related to transcription. Our data suggest that certain transcription initiation complexes, such as those that use the SSCA$_{+1}$GC motif, rely on m⁶Am for efficient expression. However, other TSS motifs do not rely as strongly on m⁶Am to achieve efficient expression. These data suggest that m⁶Am may have important roles in the transcription processes.

## CROWN-seq reveals diverse m⁶Am stoichiometry in snRNA and snoRNA

In addition to mRNAs, m⁶Am is also found on snRNAs and snoRNA (*Mauer et al., 2019*; *Koh et al., 2019*). However, the stoichiometry and dynamics of m⁶Am in these RNAs are unknown. Using

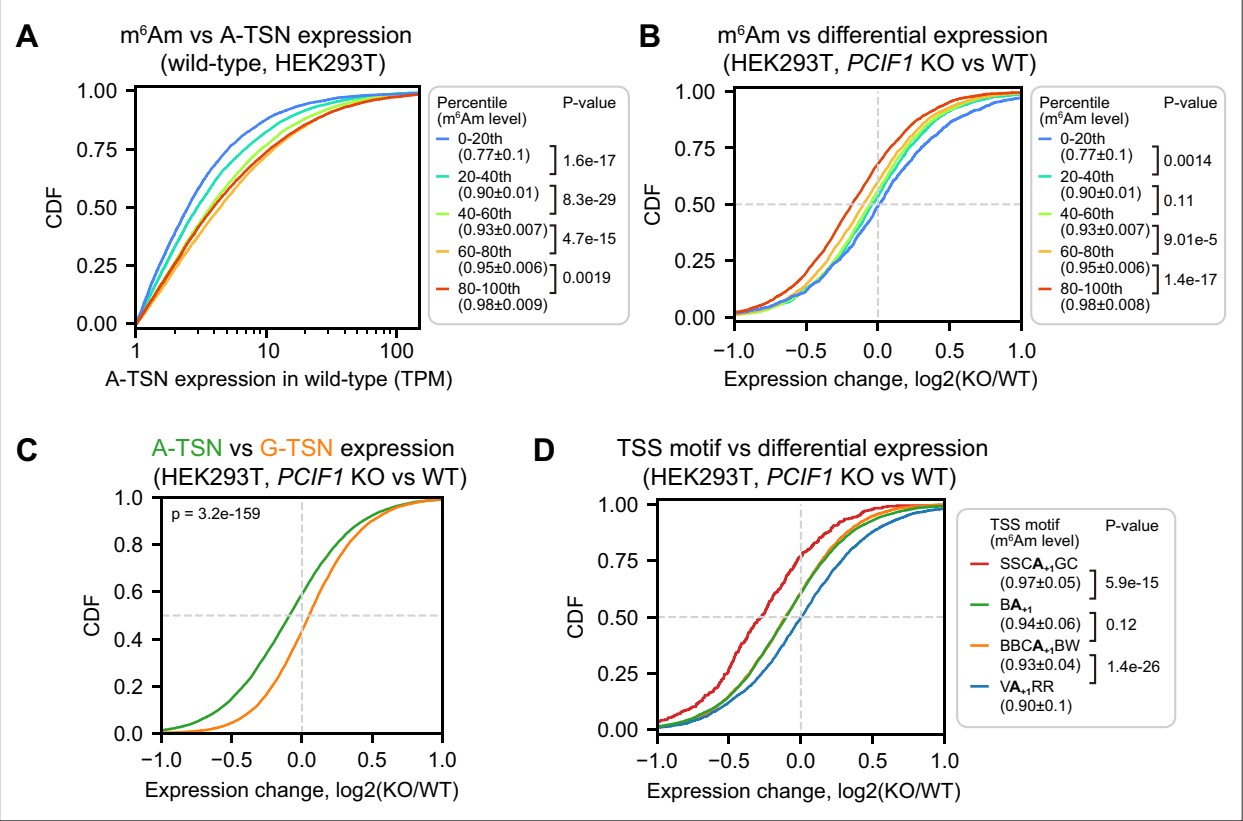

**Figure 5.** *PCIF1* knockout leads to m⁶Am and TSS motif-related A-TSN expression changes. (**A**) m⁶Am stoichiometry is positively related to A-TSN expression in wild-type HEK293T cells. In this cumulative distribution plot, the expression of each A-TSN was quantified by ReCappable-seq, for all A-TSNs in each indicated m⁶Am stoichiometry bin. A-TSNs (n=58,723) were grouped into five bins based on m⁶Am stoichiometry quantified by CROWN-seq. In total, 5125, 6962, 7991, 8368, and 8009 A-TSNs are shown in each bin (from low m⁶Am to high m⁶Am). These A-TSNs have an average TPM ≥1 in two ReCappable-seq replicates and coverage ≥50 in CROWN-seq. p-values, Student's t-test for TPM (log-transformed), two-sided. (**B**) The expression level of high m⁶Am stoichiometry A-TSNs is reduced in *PCIF1* knockout. Shown is a cumulative distribution plot of A-TSN expression change in HEK293T cells upon *PCIF1* knockout. The differential expression of A-TSN was calculated by DESeq2 (*Love et al., 2014*). Similar to (**A**), the A-TSNs were binned based on the m⁶Am stoichiometry. In total, 3269, 2272, 3218, 3813, and 3369 A-TSNs are shown in each bin (from low m⁶Am to high m⁶Am). A-TSNs with a baseMean (i.e. the average of the normalized count among replicates) ≥100 were used in the differential expression test (two replicates were used for both wild-type and *PCIF1* knockout) and coverage ≥50 reads in CROWN-seq. p-values, Student's t-test, two-sided. (**C**) Shown are cumulative distribution plots of expression changes of A-TSNs and G-TSNs after PCIF1 depletion. 14,516 A-TSNs and 9667 G-TSNs with expression levels quantified by ReCappable-seq are shown. These A-TSNs and G-TSNs have baseMean ≥ 100 during the differential expression test. p-values, Student's t-test, two-sided. (**D**) Similar to (**B**), A-TSNs that use different TSS motifs exhibit different changes in expression upon *PCIF1* knockout. In total, 352 A-TSNs using SSCA₊₁GC, 7928 A-TSNs using BA₊₁, 2958 A-TSNs using BBCA₊₁BW, and 2760 A-TSNs using VA₊₁RR are shown. These A-TSNs have baseMean ≥ 100 during differential expression test (two replicates for both wild-type and *PCIF1* knockout) and coverage ≥50 reads in CROWN-seq. p-values, Student's t-test, two-sided.

The online version of this article includes the following source data and figure supplement(s) for figure 5:

**Source data 1.** Comparing A-transcription-start nucleotide expression between wild-type and PCIF1 knockout.

**Figure supplement 1.** Corresponding to *Figure 5*.

CROWN-seq we quantified m⁶Am stoichiometry in several snRNAs, including U1, U2, U4, U4ATAC, U5, U7, U11, and U12. These snRNAs are transcribed by RNA polymerase II (*Kiss, 2004*), are capped, and use A-TSNs (*Mauer et al., 2019*). Among these snRNAs, we identified 51 m⁶Am sites, of which 29 were unannotated 5' variants located close to the annotated TSNs.

Compared with mRNA, m⁶Am in snRNA exhibited a very different distribution of stoichiometry (*Figure 2—figure supplement 1F*). First, snRNA m⁶Am sites exhibited generally low m⁶Am stoichiometry, typically below 0.3. Second, m⁶Am stoichiometry between different snRNA genes was much more variable than in mRNA (*Figure 6A*). Third, some snRNA genes show highly variable stoichiometry in different cell lines.

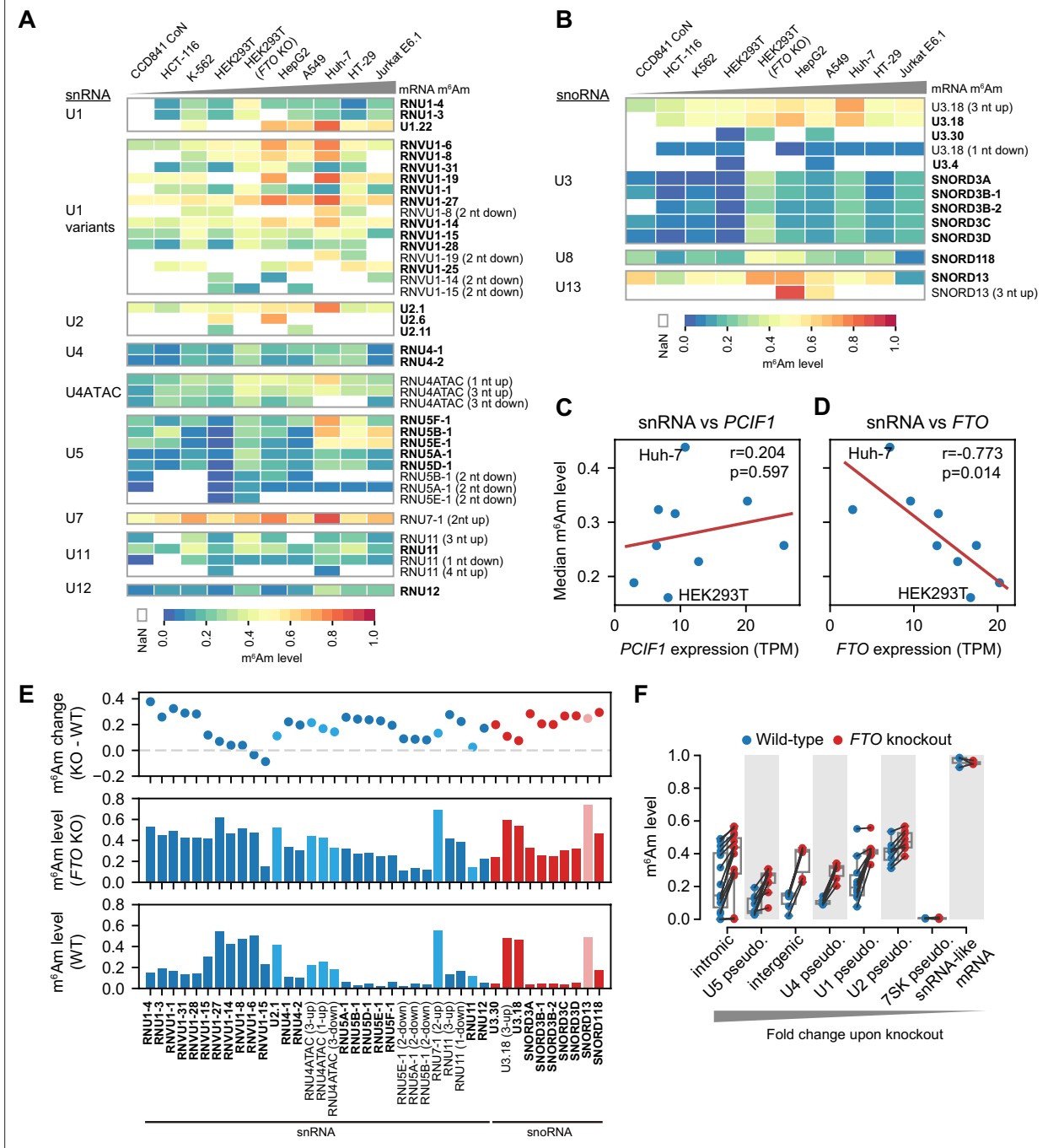

**Figure 6.** CROWN-seq reveals m⁶Am landscape in snRNA and snoRNA. (**A**) Heatmaps showing m⁶Am stoichiometry in different snRNA gene families and isoforms. Cell lines in the column are ranked by the overall mRNA m⁶Am stoichiometry. The name of each snRNA isoform is shown on the right. A-TSNs already annotated in Gencode v45 are highlighted in bold. For newly found A-TSNs, the relative distance between the new A-TSN and the nearest annotated A-TSN is shown in brackets. Note that Gencode v45 contains snRNA annotation from different databases. For example, *RNU1-4* and *U1.22* are both U1 snRNA, however, *RNU1-4* is from the HGNC database and *U1.22* is from the RFAM database. (**B**) Similar to (**A**), Heatmaps show the m⁶Am stoichiometry in different snoRNA isoforms. (**C, D**), snRNA methylation levels are not well correlated with *PCIF1* expression, but negatively correlated with *FTO* expression. The RNA expression levels of *PCIF1* and *FTO* were estimated by reading counts in CROWN-seq, which were converted into TPM to normalize the sequencing depth. Linear regressions were performed to obtain Pearson's r and p-value of the correlations. (**E**) FTO depletion leads to increased m⁶Am level (i.e. m⁶Am stoichiometry) in many kinds of snRNA and snoRNA. In this plot, the difference in m⁶Am levels between wild-type and *FTO* knockout cells is shown in the first row. The exact m⁶Am levels in *FTO* knockout and wild-type cells are shown in the second and third rows. Different kinds of snRNA and snoRNA are shown in different colors. (**F**) FTO depletion leads to increased m⁶Am stoichiometry in snRNA and snoRNA pseudogenes. In this plot, shown are the annotated pseudogenes of U1, U2, U4, U5, and 7SK, as well as the newly identified snRNA/snoRNA

*Figure 6 continued on next page*

Figure 6 continued

pseudogenes in intronic and intergenic regions. Several mRNAs exhibited 5' ends resembling snRNA pseudogenes. However, these snRNA-like mRNA 5' ends showed high and stable m⁶Am stoichiometry in both wild-type and *FTO* knockout cells.

The online version of this article includes the following source data and figure supplement(s) for figure 6:

**Source data 1.** The changes of stoichiometry of m⁶A sites close to the 5' end of mRNA upon *FTO* knockout.

**Figure supplement 1.** Corresponding to *Figure 6*.

For example, among U1 snRNA genes, *U1.22* exhibited relatively high m⁶Am levels (~0.47–0.80) in all cell lines, while *RNU1-3* and *RNU1-4* show relatively low m⁶Am levels (~0.09–0.45, *Figure 6A*). U5 snRNAs have the highest variability m⁶Am stoichiometry between cell types (*Figure 6A*). The U5 snRNA isoform *RNU5E-1* varies 31.6-fold in the m⁶Am level between HEK293T (0.0185) and Jurkat E6.1 cells (0.586). In contrast, m⁶Am in U2 and U7 snRNA are generally high (at 0.35–0.76 and 0.48–0.89, respectively) and not variable between cell lines (*Figure 6A*).

In addition to snRNA, we also examined 17 A-TSNs in C/D box snoRNA including U3, U8, and U13. These snoRNAs are involved in rRNA 2'-*O*-methylation during pre-rRNA processing (*Huang et al., 2022*). m⁶Am stoichiometry in snoRNA is highly related to snoRNA species and snoRNA isoform. For example, among different U3 snoRNA isoforms, A-TSNs of *U3.18* have much higher m⁶Am stoichiometry than others (e.g. *SNORD3A*; *Figure 6B*). These data indicate that snoRNA also has highly variable m⁶Am stoichiometry.

## FTO specifically controls m⁶Am stoichiometry in snRNA and snoRNA

We next sought to understand why m⁶Am stoichiometry is low in snRNA and snoRNA. We considered the possibility that the overall m⁶Am stoichiometry in snRNA is correlated with PCIF1 expression, as we saw with mRNA (*Figure 3B*). However, we found a poor correlation of overall m⁶Am stoichiometry in snRNA with PCIF1 expression (Pearson's $r=0.204$, p-value = 0.597, *Figure 6C*).

We next considered FTO, a highly efficient demethylase for m⁶Am in snRNA (*Mauer et al., 2019*; *Koh et al., 2019*). In contrast to PCIF1, FTO expression exhibited a strong negative correlation with snRNA methylation levels (Pearson's $r=-0.773$, p-value = 0.014, *Figure 6D*). Notably, HEK293T cells, which were tested in our previous study (*Mauer et al., 2019*), exhibited the highest FTO expression and the lowest snRNA m⁶Am stoichiometry (*Figure 6D*). Some other cell lines, such as Huh-7, have lower FTO expression and thus have relatively higher m⁶Am stoichiometry in snRNAs (*Figure 6D*).

We next wanted to determine how FTO affects m⁶Am stoichiometry in snRNAs. Using CROWN-seq on *FTO* knockout HEK293T cells, we observed prominent m⁶Am level increases in nearly all snRNA and snoRNA (*Figure 6E*). Most of the snRNA isoforms exhibited an overall increase in m⁶Am stoichiometry by ~0.2 upon *FTO* knockout. However, a subset of snRNAs were not affected by FTO depletion. For example, the *RNVU1-8* isoform has little change in m⁶Am stoichiometry. *RNVU1-8* has an unusually high m⁶Am stoichiometry at ~0.47 compared to other U1 snRNA isoforms in wild-type cells (*Figure 6E*).

Notably, FTO depletion does not increase m⁶Am levels in snRNA and snoRNA to the levels seen in mRNA (i.e. >0.9 stoichiometry). This suggests that the low m⁶Am levels in snRNA and snoRNA are not solely due to FTO-mediated demethylation. Instead, these snRNAs are likely to be inefficiently methylated by PCIF1 and are then demethylated by FTO in order to achieve their overall low m⁶Am stoichiometry.

We also found FTO demethylates m⁶Am in snRNA pseudogenes. Overall, we mapped 69 A-TSNs in annotated snRNA/snoRNA pseudogenes. These pseudogenes exhibited increased methylation upon *FTO* knockout (*Figure 6F*). We also identified 202 snRNA/snoRNA pseudogene-like transcripts. These transcripts exhibited very high similarity to the annotated snRNA/snoRNA pseudogenes, and therefore likely reflect previously unannotated pseudogenes (see Materials and methods). Upon *FTO* knockout, A-TSNs in these unannotated pseudogenes also exhibited increased m⁶Am levels (*Figure 6F*).

## FTO has minimal effects on m⁶Am and m⁶A at 5' ends of mRNA

We next asked whether FTO levels affect m⁶Am levels in mRNA. To address this question, we compared FTO RNA expression and median mRNA m⁶Am stoichiometry in all nine cell lines. This analysis shows

a weak negative correlation between FTO expression and mRNA m⁶Am (Pearson's $r=-0.239$, p-value = 0.535, *Figure 6—figure supplement 1A*).

To further assess whether FTO affects m⁶Am levels in mRNA, we quantified m⁶Am level changes in mRNA in wild-type and *FTO* knockout HEK293T cells. Overall, we observed a very small increase in mRNA m⁶Am with only a few m⁶Am sites having notably increased methylation levels upon *FTO* knockout (*Figure 6—figure supplement 1B*). Thus, only select m⁶Am sites in mRNA are efficiently demethylated by FTO.

Although CROWN-seq focuses on m⁶Am measurements, the reads in CROWN-seq can contain internal m⁶A sites if they are close to the TSN. m⁶A sites are readily detected because they do not undergo conversion with sodium nitrite. We, therefore, examined the stoichiometry of these 5'-proximal m⁶A sites in *FTO* knockout HEK293T cells. We identified internal m⁶A sites that were mapped with at least 50 reads in both wild-type and *FTO* knockout cells and had a non-conversion rate of ≥0.2 in either genotype. In total, we identified 235 high-confidence m⁶A sites which were found by both CROWN-seq and GLORI (*Liu et al., 2023*). These m⁶A sites exhibited the expected DRm⁶ACU motif (*Figure 6—figure supplement 1C*). However, these sites only showed small changes in non-conversion rates (p-value = 0.00037, paired t-test; *Figure 6—figure supplement 1D, E*). It should be noted that our conclusion about the effect of FTO on internal m⁶A is restricted to specific m⁶A sites around 5' ends since most internal m⁶A sites are not found in the 5' fragments examined in CROWN-seq.

Taken together, FTO has a strong preference for demethylating m⁶Am in snRNA, snoRNA, and their pseudogenes, compared to mRNA. FTO is a major determinant of the overall m⁶Am levels of these transcripts in different cell lines.

## Discussion

Functional studies of m⁶Am require highly accurate transcriptome-wide maps. However, m⁶Am mapping studies have relied on the assumption that each gene can be considered to have a single start nucleotide. To overcome this, we developed CROWN-seq, which maps the TSNs for all 5' transcript isoforms, and measures the exact stoichiometry of m⁶Am at all A-TSNs. CROWN-seq reveals a markedly distinct distribution of m⁶Am from what was previously recognized, largely due to inaccuracies in previous maps, and the problem with assigning each gene to a single start nucleotide. In addition, the quantitative measurements of m⁶Am in CROWN-seq show that the earlier idea that many mRNAs contain transcription-start nucleotide Am is largely incorrect since nearly all A-TSNs exhibit high stoichiometry m⁶Am. Overall, this study establishes the first quantitative, transcript isoform-specific m⁶Am map in mammalian cells. The m⁶Am maps reveal that m⁶Am is associated with increased transcript abundance, with functions of m⁶Am more correlated with transcription initiation than stability.

By selectively capturing and examining only 5' ends, CROWN-seq achieves exceptional read depth at the TSN, enabling highly accurate identification and quantification of m⁶Am. Notably, CROWN-seq is an antibody-free method and thus avoids the problem of immunoprecipitation of both m⁶Am- and m⁶A-containing fragments. This dual-specificity of antibodies creates ambiguities in m⁶Am mapping. Additionally, antibody binding cannot provide quantitative measurements of m⁶Am. In contrast, CROWN-seq uses a sodium nitrite-based chemical method for m⁶Am identification, which we show fully converts Am to Im, but leaves m⁶Am intact. Thus, the fraction of A-TSNs that contain m⁶Am or Am can be readily determined by sequencing, where all Am nucleotides are read as G. The exceptional read depth of CROWN-seq enables quantification of m⁶Am at single-nucleotide resolution, resulting in vastly more m⁶Am sites than all previous m⁶Am mapping methods combined. Although CROWN-seq involves many chemical and enzymatic steps, m⁶Am quantification by CROWN-seq is very accurate and robust, which was examined by m⁶Am standards, consistency across different technical replicates, and *PCIF1* knockout data. Notably, chemical conversion-based methods tend to have artifacts in regions with stable RNA secondary structures (*Huang et al., 2019*; *Zhang et al., 2021a*), where the nucleotides cannot efficiently interact with the chemical reagent. However, we found that CROWN-seq is very reliable even for highly structured 5' ends (*Figure 3—figure supplement 1C*), which might be due to the high accessibility of the TSN, the high stringency of the conversion steps (*Liu et al., 2023*), or RNA denaturation due to glyoxal (*Knutson et al., 2020*).

It is worth mentioning that there is no golden standard for transcription-start nucleotide (site) mapping accuracy estimation. For CROWN-seq, we first tested the mapping accuracy by in vitro transcribed RNA oligos, which shows that ~93% of the 5' ends can be mapped correctly. However, in

practice, in vitro transcription might initiate at non-specific TSSs, resulting in 5′ ends not overlapping with the desired TSSs (*Rong et al., 1998*; *Dousis et al., 2023*). Thus, the mapping accuracy can be underestimated in this assay. Since mRNA A-TSNs in the cells are known to be highly methylated by PCIF1 (*Wang et al., 2019*; *Akichika et al., 2019*), we considered that the presence or absence of m⁶Am at mapped A-TSNs can be used to assess the accuracy of TSN identification. True A-TSNs should have m⁶Am. In CROWN-seq essentially all previously annotated A-TSNs and newly found A-TSNs exhibited high non-conversion rates. These A-TSNs were well converted upon *PCIF1* knockout. This indicates very high TSN mapping accuracy, even at the many previously unannotated TSNs described here. These previously unannotated TSNs were likely missed because traditional transcription-start mapping methods and pipelines lack the sensitivity to discover them. These unannotated TSNs might have specific molecular functions. Future studies might focus on the biology of these unannotated TSNs, for example, whether these unannotated TSNs, compared to major TSNs, are associated with different mRNA processing events, such as alternative splicing.

We performed CROWN-seq in nine different cell types to understand common principles that guide m⁶Am formation in mRNA. In all cell types, m⁶Am was a very high stoichiometry modification, with some exceptions. We found a correlation between PCIF1 expression and m⁶Am stoichiometry, but even cells with very low PCIF1 expression exhibited high m⁶Am stoichiometry. The CROWN-seq data is highly consistent with recent mass spectrometry analysis of mRNA caps by us (*Wang et al., 2019*) and others (*Galloway et al., 2020*; *Akichika et al., 2019*). These mass spectrometry studies purified the entire cap structure comprising the m⁷G, the triphosphate linker, and the first nucleotide. In these studies, m⁷G-ppp-m⁶Am was highly prevalent while m⁷G-ppp-Am abundance was typically 1/10 as m⁷G-ppp-m⁶Am in nearly all cell lines (*Wang et al., 2019*; *Galloway et al., 2020*). This mass spectrometry data was the first suggestion that transcription-start nucleotide Am was not a prevalent modification in mRNA, as had been suggested by early chromatography studies (*Wei et al., 1975*). We suspect that the high levels of Am seen in these early analyses of mRNA can be explained by contaminating snRNA or rRNA fragments, which are highly difficult to remove, even with multiple rounds of poly(A) purification (*Legrand et al., 2017*). It remains possible that there are cell types or cellular contexts that remain to be discovered with low m⁶Am (i.e. high Am) levels. However, it is clear that high m⁶Am stoichiometry is a general feature of most or all cell types in this study.

The initial m⁶Am maps relied on published TSN annotations. In the first m⁶Am map, annotations were based on FANTOM5 (*Noguchi et al., 2017*), which primarily uses CAGE datasets to define the start nucleotide. However, these annotations selected a single start nucleotide even if multiple TSS signals from CAGE peaks were detected for a gene (*Akichika et al., 2019*; *Boulias et al., 2019*). It should be noted that some genes may primarily use m⁶Am for all 5′ transcript isoforms. These genes would therefore have a high m⁶Am gene index. Genes with a high m⁶Am gene index are likely to be preferentially affected by PCIF1 depletion or pathways that affect m⁶Am.

Based on the small range of m⁶Am stoichiometry in A-initiated mRNAs, it is unlikely that the variability in stoichiometry has functional significance for most mRNAs. Instead, our data suggest that mRNAs initiate with either Gm, Cm, Um, or Am, where Am is highly m⁶Am modified. mRNAs that initiate with m⁶Am may have shared regulatory mechanisms that distinguish them from mRNAs that initiate with Gm, Cm, and Um. Additionally, genes that primarily initiate with m⁶Am, either because they have only one major transcription-start site, or because all their transcription-start nucleotides are A, would be highly influenced by m⁶Am-dependent regulatory mechanisms. Currently, cellular pathways that target m⁶Am-initiated mRNAs are not well known.

Our study revealed a link between m⁶Am and transcription. This effect was detectable because of the highly quantitative nature of m⁶Am measurement in CROWN-seq. Although all A-TSNs show high stoichiometry, there are differences in the overall m⁶Am stoichiometry between transcripts, for example ~0.85 stoichiometry vs. 0.95 stoichiometry which can readily be detected by CROWN-seq. We found that these differences are often related to the specific TSS motif. For example, the Inr-like SSCA$_{+1}$GC TSS motif was associated with the highest m⁶Am stoichiometry, while transcripts using the VA$_{+1}$RR TSS motif exhibited relatively lower m⁶Am stoichiometry. This finding highlights that the major role of m⁶Am might be linked to transcription regulation, which is supported by a recent study by .*An et al., 2024*.

We then examined the effects of PCIF1 depletion on m⁶Am transcript abundance. We found that transcripts with higher methylation in wild-type cells tend to have a larger reduction in RNA expression

level upon *PCIF1* knockout. Further analysis showed that transcripts that use the SSCA$_{+1}$GC TSS motif exhibited significantly reduced expression in *PCIF1* knockout cells. In contrast, transcripts that use the VA$_{+1}$RR TSS motif were largely unaffected. Notably, transcripts normally have small differences in methylation (i.e. methylation level at 0.9 vs 0.98). Thus, m⁶Am is unlikely to be a general stabilization mark in mRNA since it only affects transcripts based on promoter sequences. Instead, these different stoichiometries of m⁶Am are likely to be the consequence of different transcription mechanisms. Thus, it will be important to assess how these different transcription mechanisms use m⁶Am for gene expression.

PCIF1 is known to be associated with RNA polymerase II and is recruited to promoter regions during transcription (*Sugita et al., 2021*). Thus, PCIF1 is ideally positioned to regulate transcription processes. It is interesting to speculate that m⁶Am may provide a mark that enhances subsequent elongation and thus maintains high overall expression for transcription initiation complexes that assemble on the SSCA$_{+1}$GC TSS motif. Other transcription initiation complexes, such as those using the VA$_{+1}$RR TSS motif, may not need this mechanism. However, our data cannot provide further details on whether the loss of m⁶Am is related to exact mechanisms such as transcription initiation selection, elongation, or premature termination. Notably, the recent study by *An et al., 2024* suggested that the loss of m⁶Am is related to enhanced premature termination and therefore leads to reduced RNA 5′ end expression. An et al. proposed that m⁶Am can sequester PCF11, an m⁶Am reader, and thereby promote transcription by reducing premature transcription termination (*An et al., 2024*). However, it is still unclear whether the transcripts from the SSCA$_{+1}$GC TSS motif are indeed more preferentially bound by PCF11. To better understand how PCIF1 regulates transcription, assays with transcription-start nucleotide resolution will be required.

Although m⁶Am and m⁶A are chemically similar, these two modifications appear to have very different biological functions. It is well known that m⁶A is a mark for RNA instability through the recruitment of YTHDF proteins (*Wang et al., 2014*). However, we find no correlation between m⁶Am and RNA instability. Additionally, our previous YTHDF1, YTHDF2, and YTHDF3 iCLIP studies did not show binding at mRNAs 5′ ends (*Patil et al., 2016*), which suggests that YTHDF proteins do not bind m⁶Am. Thus, specific m⁶Am-binding proteins might enable its unique functions in transcription.

Although most studies of m⁶Am and PCIF1 focus on mRNAs, we find that m⁶Am in snRNAs exhibit substantially higher variability and regulation than that in mRNA. Early biochemical studies of snRNA composition demonstrated that the first nucleotide was generally Am in all Pol II-derived snRNAs (*Mauer et al., 2019*). CROWN-seq generally supports this finding since most snRNAs have low m⁶Am stoichiometry. However, the previous study mainly focused on HEK293T cells (*Mauer et al., 2019*), which have very low m⁶Am in snRNA. In this study, nine different cell types were sequenced. These cell lines showed highly variable m⁶Am in snRNA. In some cases, several snRNAs can reach m⁶Am stoichiometry up to 0.70–0.89. These data raise the possibility that m⁶Am may affect snRNA functions, such as splicing and gene transcription (*Koh et al., 2019*; *Mimoso and Adelman, 2023*), and *PCIF1* knockout phenotypes may be due to altered snRNA.

Notably, m⁶Am in snRNA is highly regulated by FTO, which is consistent with our earlier findings (*Mauer et al., 2019*). However, the previous study did not have transcript isoform level resolution in analyzing the effect of FTO demethylation. With CROWN-seq, we find that FTO has markedly different effects on different snRNAs, where some snRNAs appear highly demethylated by FTO while others are insensitive to FTO. Some snoRNA, and snRNA/snoRNA pseudogenes are also demethylated by FTO. Notably, FTO depletion affects numerous aspects of cell function (*Mauer and Jaffrey, 2018*). Our results thus raise the possibility that FTO-depletion phenotypes may result from increased m⁶Am levels in snRNAs, snoRNAs, or their pseudogenes.

## Limitations of the study

One limitation of CROWN-seq is that it can be difficult to align sequencing reads to the genome. Unlike normal reads, which contain A, G, C, and U, most reads in CROWN-seq comprise only G, C, and U due to the conversion of A's. This makes it difficult to align reads to highly similar genes, such as snRNA isoforms and pseudogenes which have very similar 5′ ends. For this reason, only a small fraction of reads from snRNA and pseudogenes were uniquely mapped to one genomic location and were used in this analysis. To better understand m⁶Am in these 5′ ends with similar sequences, future optimization is desired to increase the read lengths, which can help distinguish these sequences from

each other. This requires technical innovations in reducing RNA fragmentation during sodium nitrite conversion, which comes from acid-catalyzed depurination and backbone cleavage (*Mahdavi-Amiri et al., 2021*).

In this study, we quantified m$^6$Am in nine different cell lines, which cover a wide range of PCIF1 expression levels. Although we found high m$^6$Am stoichiometries in all cell lines, it is possible that some cells or tissues have more variable m$^6$Am levels. In our previous study, mass spectrometry showed that the CCRF-SB cell line has relatively low m$^6$Am stoichiometry at ~67.6% (*Wang et al., 2019*). However, these cells exhibit very slow growth as reported previously (*Wang et al., 2019*). As a result, we were unable to obtain enough mRNA needed for CROWN-seq. Future CROWN-seq studies may lead to the identification of cell types or contexts with dynamic m$^6$Am landscapes.

The last limitation of this study is that the focus of this study was to quantify m$^6$Am and predict potential functions using *PCIF1* knockout cells. However, it is possible that PCIF1 has non-catalytic functions that may contribute to the *PCIF1* knockout phenotype. Future experiments using catalytic-dead PCIF1 can be useful to distinguish between the catalytic and non-catalytic functions of PCIF1.

# Materials and methods

## Key resources table

| Reagent type (species) or resource | Designation | Source or reference | Identifiers | Additional information |
|---|---|---|---|---|
| Gene (*Homo sapiens*) | *PCIF1* | Enesmbl | ENSG00000100982 | |
| Gene (*H. sapiens*) | *FTO* | Enesmbl | ENSG00000140718 | |
| Cell line (*H. sapiens*) | HEK293T | ATCC | CRL-3216 | |
| Cell line (*H. sapiens*) | HEK293T, *PCIF1* knockout | *Boulias et al., 2019* | | |
| Cell line (*H. sapiens*) | HEK293T, *FTO* knockout | *Mauer et al., 2019* | | |
| Cell line (*H. sapiens*) | A549 | ATCC | CCL-185 | |
| Cell line (*H. sapiens*) | A549, *PCIF1* knockout | This study | | |
| Cell line (*H. sapiens*) | HepG2 | ATCC | HB-8065 | |
| Cell line (*H. sapiens*) | Huh-7 | ThermoFisher | huh 7 Cells | |
| Cell line (*H. sapiens*) | Jurkat E6.1 | ATCC | TIB-152 | |
| Cell line (*H. sapiens*) | K-562 | ATCC | CCL-243 | |
| Cell line (*H. sapiens*) | CCD841 CoN | ATCC | CRL-1790 | |
| Cell line (*H. sapiens*) | HCT-116 | ATCC | CCL-247 | |
| Cell line (*H. sapiens*) | HT-29 | ATCC | HTB-38 | |
| Sequence-based reagent | ReCappable-seq 5' adapter (11 N) | IDT | RNA adapter | rCrCrUrArCrArCrGrArCrGrCrUrCrUrUrCrCr GrArUrCrUrNrNrNrNrNrNrNrNrNrNrNrArUrArU |
| Sequence-based reagent | ReCappable-seq 3' adapter | IDT | DNA adapter | /5rApp/WWAGATCGGAAGAGCACACGTC/3ddC/ |
| Sequence-based reagent | CROWN-seq 5' adapter (8 N) | IDT | RNA adapter | rCrCrUrArCrArCrGrArCrGrCrUrCrUrUr CrCrGrArUrCrUrNrNrNrNrNrNrNrNrArUrArU |
| Sequence-based reagent | CROWN-seq 5' adapter (11 N) | IDT | RNA adapter | rCrCrUrArCrArCrGrArCrGrCrUrCrUrUrCr CrGrArUrCrUrNrNrNrNrNrNrNrNrNrNrNrArUrArU |
| Sequence-based reagent | CROWN-seq 3' adapter | IDT | RNA adapter | /5'rApp/AGATCGGAAGAGCACACGTCTGAACT CCAGTCACAAAAAAAAAAAAAAAACCCCCCCCCCA AAAAAAAAAAAAAA/3AmMO/ |
| Sequence-based reagent | ReCappable-seq/ CROWN-seq RT primer | IDT | RT primer | GACGTGTGCTCTTCCGATCT |

*Continued on next page*

*Continued*

| Reagent type (species) or resource | Designation | Source or reference | Identifiers | Additional information |
|---|---|---|---|---|
| Sequence-based reagent | GLORI 5' adapter (11 N) | IDT | RNA adapter | rCrCrUrArCrArCrGrArCrGrCrUrCrUrUrCrCr GrArUrCrUrNrNrNrNrNrNrNrNrNrNrArUrArU |
| Sequence-based reagent | GLORI 3' adapter | IDT | DNA adapter | /5rApp/AGATCGGAAGAGCACACGTC/3AmMO/ |
| Sequence-based reagent | GLORI RT primer | IDT | RT primer | GACGTGTGCTCTTCCGATCT |
| Sequence-based reagent | PCIF1_qPCR_F | IDT | qPCR primer | GGAGAATCGTCCCTACTACTT |
| Sequence-based reagent | PCIF1_qPCR_R | IDT | qPCR primer | GCTTTCTGGGCTTGTTCT |
| Sequence-based reagent | GAPDH_qPCR_F | IDT | qPCR primer | GTGGACCTGACCTGCCGTCT |
| Sequence-based reagent | GAPDH_qPCR_R | IDT | qPCR primer | GGAGGAGTGGGTGTCGCTGT |
| Software, algorithm | HISAT2 | *Kim et al., 2019* | RRID:SCR_015530 | v2.2.1 |
| Software, algorithm | UMI-tools | *Smith et al., 2017* | RRID:SCR_017048 | v1.1.1 |
| Software, algorithm | BEDtools | *Quinlan and Hall, 2010* | RRID:SCR_006646 | v2.27.1 |
| Software, algorithm | SAMtools | *Li et al., 2009* | RRID:SCR_002105 | v1.16.1 |
| Software, algorithm | Python3 | Python | RRID:SCR_008394 | v3.8.7 |
| Software, algorithm | R | R | RRID:SCR_001905 | v4.2.2 |
| Software, algorithm | numpy | PyPI | RRID:SCR_008633 | v1.23.5 |
| Software, algorithm | pandas | PyPI | RRID:SCR_018214 | v1.5.2 |
| Software, algorithm | scipy | PyPI | RRID:SCR_008058 | v1.93 |
| Software, algorithm | pysam | *Li et al., 2009* | RRID:SCR_021017 | v0.19.1 |
| Software, algorithm | DESeq2 | *Love et al., 2014* | RRID:SCR_015687 | v1.38.1 |
| Software, algorithm | RUVSeq | *Risso et al., 2014* | RRID:SCR_006263 | v1.38.0 |
| Software, algorithm | GLORI analysis pipeline | This paper | | v1.0; https://github.com/jhfoxliu/GLORI_pipeline |
| Software, algorithm | ReCappble-seq analysis pipeline | This paper | | v1.0; https://github.com/jhfoxliu/ReCappable-seq |
| Software, algorithm | CROWN-seq analysis pipeline | This paper | | v1.0; https://github.com/jhfoxliu/CROWN-seq |

## Experimental model and subject details

### Cell lines

HEK293T, A549, Jurkat E6.1, HCT-116, HT-29, CCD841 CoN, K562, and HepG2 were purchased from ATCC (American Type Culture Collection). Huh-7 was purchased from Thermo Fisher. *PCIF1* knockout, and *FTO* knockout cells were generated by CRISPR knockout, validated by Western blots and m⁶Am TLC. The identities of the cell lines were authenticated by STR profiling. No mycoplasma contamination was detected.

HEK293T (wild-type, *PCIF1* knockout, and *FTO* knockout cells), A549 (wild-type and *PCIF1* knockout), HCT-116, Huh-7, and HT-29 cells were maintained in DMEM (Gibco #11995065). HepG2 and CCD841 CoN cells were maintained in EMEM (ATCC #30–2003). K562 and Jurkat E6.1 cells were maintained in RPMI1640 (Gibco #11875093). All media was supplemented with 10% FBS and 1 X penicillin-streptomycin (Gibco #15140148). Cells were grown at 37 °C with 5% $CO_2$. We followed the instructions from the manufacturer to maintain the cells.

## Methods details

### RNA extraction and mRNA purification

Cellular total RNA in TRIzol LS (Thermo Fisher #10296028) was extracted by Direct-zol RNA Miniprep kit (Zymo #R2070) or by Phenol Chloroform extraction. mRNA was purified by NEBNext Oligo d(T)25 Beads (NEB #E7499) or Dynabeads Oligo (dT)25 (Ambion #61002) based on mRNA purification from total RNA protocol of Dynabeads Oligo (dT)25 (Ambion #61002).

### m⁶Am standard preparation

We used in vitro transcription to prepare m⁷G capped m⁶Am- or Am-initiated transcripts, which are based on HiScribe T7 mRNA Kit with CleanCap Reagent AG (NEB #E2080S). We first obtained DNA templates from IDT gBlock. In total, five DNA templates which are identical expect for the 6-nt long barcode 42-nt downstream to the TSS were used. The DNA templates contain 5'-TAATACGACTCACTATAAG-3' T7 promoter for in vitro transcription. We used CleanCap Reagent AG (3' OMe) (TriLink #N-7413), which is included in NEB #E2080S, to generate m⁷G-ppp-Am modified transcripts. We used CleanCap Reagent M6 (TriLink #N-7453) to generate the m⁷G-ppp-m⁶Am modified transcripts. The RNAs made by in vitro transcription were DNase I treated, purified, and then quantified by both Agilent TapeStation (RNA high sensitivity assay). We then mixed the Am and m⁶Am modified oligos to generate m⁶Am standards with expected m⁶Am stoichiometry at 0%, 25%, 50%, 75%, and 100% m⁶Am stoichiometry. Notably, the guaranteed purity of the CleanCap Reagent M6 is >95%. The CleanCap Reagent M6 can contain m⁷G-ppp-AmG analog, which results in the reduced non-conversion rate in CROWN-seq.

### Genomic assembly and annotations

The genomic sequence and annotations of Gencode v45 primary assembly were used in this study.

### GLORI experiment

To validate whether sodium nitrite conversion can convert Am into Im, we spiked ~0.01 ng Am transcripts (ERCC-00057–1-TCGTCG) into ~250 ng poly(A) selected mRNA for GLORI assay. Ligation-based GLORI protocol was used in this study. Notably, the Am transcripts were decapped by mRNA Decapping Enzyme (NEB #M0608S) in advance. We first fragmentized the input RNA into ~200 nt long fragments (NEBNext Magnesium RNA Fragmentation Module (NEB #E6150S), 94 °C, 2 min). The fragmentized RNAs were then A-to-I converted based on the GLORI protocol (*Liu et al., 2023*): we converted the glyoxal-protected RNA by 750 mM NaNO₂ at 16 °C for 8 hr and 4 °C overnight. The RNA was then deprotected in a deprotection buffer at 95 °C for 10 min. The deprotected RNA was then T4 PNK (NEB #M0210S) treated and processed to ligation-based small RNA-seq library preparation (*Vigneault et al., 2012*). Notably, the 5' adapter in library preparation contains an 11 nt UMI sequence.

### GLORI data processing

GLORI libraries were analyzed based on a modified mRNA bisulfite sequencing pipeline (*Huang et al., 2019*). The first 10 bases in GLORI libraries made with eCLIP protocol were first extracted by a customized script. GLORI reads were first quality trimmed by Cutadapt (*Martin, 2011*). For the GLORI library generated by eCLIP protocol, the parameters are `--max-n 0 --trimmed-only -a AGATCGGAAGAGCGTCGTG -e 0.1 -q 30 m 40 --trim-n`; for GLORI library prepared by ligation-based protocol generated in this study, the parameters are `-m 32 j 4 -q 20 -e 0.25 -a AGATCGGAAGAGCACACGTC -A ATATNNNNNNNNNNNAGATCGGAAGAGCGTCGTG`. After pre-processing, the reads were firstly A-to-G converted and aligned to A-to-G (positive strand) and T-to-C (negative strand) converted reference genome and transcriptome by Hisat2-2.1.0 (*Kim et al., 2019*). Parameters in alignment: `-k 5 -fr -rna-strandness FR -no-temp-splicesite -no-mixed`. Only unique alignments were used. After alignment, the base information in sequences was restored so that m⁶Am signals can be reflected by the A-to-G mismatches. No further transcriptome alignment was performed on the unmapped reads. After alignment, a customized script based on Pysam (*Li et al., 2009*) was used to pileup every single base to obtain the A, C, G, and U counts. Every single base was assigned to a transcript isoform if possible (order: `mRNA > lncRNA > functional RNAs`

> `pseudogenes`). Non-conversion rate is defined as the number of A counts against the sum of A count and G count. Filters were applied to obtain high-quality non-converted A (m⁶A/m⁶Am) signals in a gene-specific manner: (1) only genes with at least 1000 counts were analyzed; (2) gene-specific non-conversion rates were computed for Binomial test on the frequency of non-conversion and sites with Binomial test p-value <0.05 were used; (3) reads with more than three non-converted As were considered as noise and discarded; (4) sites with more than 5% signals were discarded due to the site may fall in a conversion-resistant region; (5) Only sites with no less than 20 reads covered and non-conversion rates over 0.1 were considered as m⁶A/m⁶Am sites. (6) Non-conversion rates of the same site from different replicates were averaged. Details of this pipeline can be found at https://github.com/jhfoxliu/GLORI_pipeline (**Liu, 2023**).

## ReCappable-seq library preparation

A modified ReCappable-seq protocol (**Yan et al., 2022**) was developed to reduce background, reduce material loss, and increase the utility of mapped reads. Several steps of library construction are now performed while the 5′ desthiobiotinylated cap is bound to streptavidin beads. This reduces the opportunity for carry-through of random fragmentation products to occur that would previously result in non-cap-derived 5′ ends to be ligated. Next, 5′ adapters with unique molecular indexes (UMIs) are used to permit robust PCR duplicate removal. Finally, ~160 spike-in mRNAs from SIRV-ERCC Spike-in mixture (Lexogen #051.03) with single defined 5′ termini are used, which are used during analysis to build a dynamic thresholding pipeline that exclude false positive start sites. A complete step-by-step protocol as performed here are available at https://github.com/jhfoxliu/ReCappable-seq (**Liu, 2025a**).

5 µg total RNA was used as input for all experiments. RNA was denatured at 65 °C for 2 min before reaction mixes were added. First, 5′-phosphorylated RNAs were dephosphorylated using 25 U Quick CIP (NEB #M0525L) in a 50 µL reaction for 30 min at 37 °C. The reaction was cleaned using a Zymo RCC-5 column following the manufacturer's>200 nt protocol and eluted with 20 µL water. m⁷G capped RNAs were then decapped using 200 U yDcpS (NEB #M0463S) for 1 hr at 37 °C. This unique decapping enzyme liberates m⁷GMP, resulting in mRNAs with a 5′-diphosphate. The reaction was cleaned and eluted as before. Next, the 5′-diphosphorylated mRNAs were recapped with desthio-biotin-GTP (DTB-GTP) using vaccinia capping enzyme (5 µL VCE buffer, 0.5 µL inorganic pyrophos-phatase (NEB #M0361S), 5 µL DTB-GTP (5 mM; NEB #N0761S), 50 U VCE (#M2080S)) for 45 min at 37 °C. The reaction was clean as before, however, a total of our washes were performed to ensure the complete removal of excess DTB-GTP. RNA was then fragmented by incubating at 95 °C for 2.5 min in a 25 µL reaction containing 100 mM Tris-HCl pH 8.0 and 2 mM MgCl₂. Fragmented RNA was placed on ice and brought to 30 µL with water. Streptavidin beads (NEB #S1421S) were washed in a high-salt wash buffer (10 mM Tris-HCl pH 7.5, 2 M NaCl, 1 mM EDTA) and resuspended in the high salt buffer at 4 mg/mL. 30 µL beads were added to 30 µL fragmented RNA and incubated for 45 min at room temperature with agitation. Beads were washed twice in a high-salt buffer, twice in a lower salt buffer (10 mM Tris-HCl pH 7.5, 250 mM NaCl, 1 mM EDTA), and twice in PNK wash buffer (20 mM Tris-HCl pH 7.5, 10 mM MgCl₂, 0.2% Tween). Beads were next resuspended in 40 µL PNK reaction mix 8 µL 5 X pH 6.5 PNK buffer (350 mM Tris-HCl pH 6.5, 50 mM MgCl₂, 5 mM DTT), 1 µL T4 PNK (NEB #M0201S), 1 µL RNaseOUT (Thermo Fisher #10777019) and incubated at 37 °C for 30 min with agitation to remove 3′ phosphates resulting from the fragmentation. Beads were washed once in PNK wash, once in the high salt wash, then twice again in PNK wash. Next, a 3′ adapter was added to RNA by resuspending beads in 40 µL 3′ ligation reaction mix (4 µL T4 RNA ligase buffer, 2 µL T4 RNA ligase 2 truncated KQ (NEB #M0373L), 1 µL RNaseOUT, 2 µL L7 adapter (20 µM stock), 16 µL of 50% PEG-8000) and incubated at 25 °C for 2 hr. The beads were washed once in PNK wash, once in high-salt wash, twice in lower salt wash, then resuspended in 30 µL lower salt wash containing 1 mM biotin (ThermoFisher #B20656) to elute DTB-capped RNA fragments. The eluted RNA was cleaned by ethanol-AMPure XP (1.8 volumes AMPure XP, then 1.5 volumes 100% ethanol). To increase stringency, the streptavidin bead enrichment was repeated omitting enzymatic steps and instead washing three times with high salt and then three times with lower salt wash, and the eluate was cleaned again by ethanol-AMPure XP. The DTB-GTP cap was removed using 0.5 U/µl RppH (NEB #M0356S) in 1 X Ther-moPol buffer (NEB #M0356S) and incubating at 37 °C for 1 hr. The resulting 5′-monophosphate RNA fragments were purified by ethanol-AMPure XP. 30 pmol of a 5′ adapter was ligated for 3 hr at 25 °C with 2 U/µL T4 RNA ligase 1 (NEB #M0437M). This RNA adapter contains an 11 nt UMI followed by a

fixed sequence (AUAU) at its 3' end. The UMI allows robust duplicate removal, and the fixed sequence provides an anchor point to correctly identify the first nucleotide of the mRNA. The ligation reaction was inactivated by heating at 65 °C for 10 min and then immediately used in a reverse transcription reaction. 3 pmol of ReCappable-seq RT primer was annealed to the 3' adapter of RNA fragments by heating to 65 °C for 5 min and cooling to 25 °C at a rate of 0.1 °C/s. Reverse transcription was carried out at 55 °C for 45 min in a reaction containing 0.5 mM dNTPs, 5 mM DTT, 20 U RNaseOUT, 50 mM Tris-HCl pH 8.3, 75 mM KCl, and 300 U SuperScript III (Thermo Fisher #18080044). Following heat inactivation, the reaction was cleaned using ethanol-AMPure XP and cDNA was resuspended in 21 µL. The final PCR was performed using 8 µL cDNA in a 40 µL reaction containing 1 X Phusion HF master mix (NEB #M0531L) and 4 µL each of a unique i5 and i7 barcoded primer combination for each sample (NEB #E7600S). Cycling conditions were typically 98 °C 2 minutes, then 11–13 cycles of 98 °C 15 s, 65 °C 30 s, 72 °C 30 s, with a final 5 min 72 °C extension. The optimal number of cycles for each library was determined by performing a set of test cycles using 1 µL cDNA in a 20 µL reaction. PCR libraries were purified with 2 rounds of bead clean-up using 0.9 X volume SPRIselect beads. Libraries were pooled at equimolar concentrations and sequenced in paired-end mode with 50–150 bp reads depending on the library on either an Illumina NovaSeq, NextSeq, or HiSeq (please refer to GEO accession for specific details for each library).

## ReCappable-seq analysis

The beginning of read 1 is the UMI plus an ATAT spacer sequence, and the nucleotide directly following this is the TSS. Reads were first filtered to identify pairs with the correct UMI +ATAT sequence, then the UMI was added to FASTQ headers using UMI-tools v1.1.1 (*Smith et al., 2017*). ATAT sequence discarded. Adapters were trimmed using Cutadapt v3.4 (*Martin, 2011*). Next, reads mapping to ribosomal RNA and small non-coding RNAs were filtered away by aligning to these sequences using bowtie2 v2.4.2 (*Langmead and Salzberg, 2012*). Reads were then aligned to GRCh38 and m⁶Am standard sequences using HISAT2 (*Kim et al., 2019*). The alignment results were dedupli-cated by UMI-tools (`--paired --chimeric-pairs=discard --unpaired-reads=discard --method=unique`). Only reads without 5' softclipping were used. A customized script based on Pysam (*Li et al., 2009*) was used to extract the 5' ends from the BAM file. To annotate the sites by a gene, the 5' ends were firstly annotated by the nearest TSS within the 100 bp region. If multiple annotations were found, the annotation was selected by the priority of `snRNA > snoRNA > mRNA > lncRNA > others`. BEDtools v2.27.1 (*Quinlan and Hall, 2010*) was used to find the nearest annota-tion. To more accurately estimate the expression levels of each TSN, we normalized the read counts using the 'RUVg' function in RUVSeq pacakge (*Risso et al., 2014*). To calculate the expression levels of TSNs in wide-type cells, we calculated the TPM values based on the normalized read counts. To calculate the differential expression between wild-type and *PCIF1* knockout cells, the normalized read counts were proceeded by DESeq2 (*Love et al., 2014*).

## The comparison of TSS mapping methods

Currently, there are several types of TSS mapping methods. CAGE (*Murata et al., 2014*) and TSS-seq (*Yamashita et al., 2011*) are the two most popular methods being used.

CAGE was tested to have the highest precision and sensitivity over other TSS mapping methods (*Adiconis et al., 2018*), except for ReCappable-seq. However, CAGE has two limitations. First, CAGE relies on template switching. Template switching is a process in that reverse transcriptase can 'jump' onto a template switching oligo, which contains an adapter sequence when the reverse transcriptase reaches the end of the RNA template. Template switching is very convenient in producing full-length cDNA without ligating adapter. However, template switching is not precise for transcription-start nucleotide identification, because template switching can introduce non-template bases (normally C's) into the cDNA between the template and the adapter. It is very difficult to completely remove the non-template bases in the CAGE library because the number of incorporated non-template bases is uncertain (*Tang et al., 2013*). As a result, compared with TSS-seq and ReCappable-seq, CAGE can mistakenly assign TSSs within the same CAGE peak (see *Figure 1—figure supplement 1A, B*). Second, the most widely used CAGE protocol (*Murata et al., 2014*) contains an oxidation step, which results in massive indels and mutations in the cDNA. These indels and mutations can result in inaccurate alignments. Third, 'strand invasion' can cause TSS artifacts in CAGE (*Tang et al., 2013*).

Strand invasion is the process that reverse transcriptase mistakenly terminates and switches onto the template switching oligo before reaching the end of a template. Strand invasion can result in false positive TSSs in internal RNA positions.

TSS-seq is another available method in TSS mapping. TSS-seq relies on several enzymatic steps to remove non-m$^7$G-capped 5′ end backgrounds in the sample. After removing undesired 5′ ends, the m$^7$G cap is released and a 5′ adapter is ligated to the RNA 5′ ends. In theory, this procedure can result in precise 5′ end maps. However, tested by *Adiconis et al., 2018*, TSS-seq exhibited low precision, sensitivity, and accuracy in TSS mapping. The low performance of TSS-seq is due to the incomplete removal of the non-m$^7$G-capped 5′ end backgrounds.

ReCappable-seq (*Yan et al., 2022*) can be considered as an improved TSS-seq. Recappable-seq overcomes the 5′ end background clean-up issue. In ReCappable-seq, the m$^7$G caps of RNA polymerase II transcribed RNA is replaced by 3′-Desthiobiotin-G caps. The recapped RNAs can thus be enriched on streptavidin beads. During high-stringency washing, the 5′ end background can be completely removed. Thus, ReCappable-seq exhibited extremely high specificity in mapping transcription-start nucleotides.

## CROWN-seq library preparation

CROWN-seq uses the glyoxal-based guanosine protection protocol from GLORI (*Liu et al., 2023*) and a TSN enrichment protocol that is modified from ReCappable-seq (*Yan et al., 2022*). In CROWN-seq, glyoxal protection is very important to prevent both internal G's from being converted into xanthosine, which can interrupt base pairing and cause mutations during reverse transcription (*Mair et al., 2022*). Because N7-methyl does not interrupt the interaction between glyoxal and N1 and N2 positions of guanosines, glyoxal protection is also very useful to prevent m$^7$G from being converted, which can help 5′ end enrichment. After glyoxal protection, A bases are deaminated into inosines by sodium nitrite. After deamination, the 5′ end RNA fragments with a m$^7$G cap were enriched by ReCappable-seq workflow, where the m$^7$G caps were replaced by a 5′ desthio-biotinylated cap for enrichment by streptavidin beads. 3′ adapter and 5′ adapter (with unique molecular indexes [UMIs]) were ligated to the enriched 5′ RNA fragments, so that the library can be made by reverse transcription followed by indexing PCR. Detailed workflow is described below.

Conversion. 0.8–2.5 µg oligo(dT) selected RNA was used as input. RNA was first diluted in 14 µl water. To perform glyoxal protection, 6 µl 8.8 M glyoxal and 20 µl DMSO were then added to the diluted RNA and well mixed. The 40 µl mix was first incubated at 50 °C for 30 min, then 10 µl boric acid was added to the mix. The 50 µl mix was then incubated for an additional 30 min at 50 °C. After protection, the 50 µl protected RNA was mixed with 50 µl deamination buffer (25 µl 1500 mM NaNO$_2$, 4 µl 500 mM MES, pH 6.0, 10 µl 8.8 M glyoxal, and 11 µl water). The deamination reaction was performed at 16 °C for 8 hr. After deamination, the RNA was recovered by ethanol precipitation. To remove the glyoxal adduct from the RNA, the RNA pallet was dissolved in 50 µl deprotection buffer (500 mM TEAA pH = 8.6, 47.5% deionized formamide) and was incubated at 95 °C for 10 min. After incubation, the reaction was brought to 250 µl with water. Converted RNA was purified by ethanol precipitation and eluted in 39 µl water for 5′ end enrichment. The converted RNA was stored at –80 °C before 5′ end enrichment.

Recapping. To eliminate the contamination of RNA with 5′-triphosphate and 5′-monophosphate, 5 µl 10 X CutSmart buffer, 5 µl Quick CIP (5 U/µl) (NEB #M0525L), and 1 µl SUPERase·In RNase inhibitor (Thermo Fisher #AM2696) were added to the 39 µl converted RNA to set up a dephosphorylation reaction. The dephosphorylation reaction was performed at 37 °C for 30 min. The reaction was cleaned up using Zymo RCC-5 column and the RNA was eluted in 42 µl water. To decap the m$^7$G capped RNA, a 50 µl decapping reaction was set up by adding 5 µl 10 X yDcpS buffer, 2 µl (200 U) yDcpS (NEB #M0463S), and 1 µl SUPERase·In to the 42 µl dephosphorylated RNA. The decapping reaction was performed at 37 °C for 1 hr. This unique decapping enzyme liberates m$^7$GMP, resulting in mRNAs with a 5′-diphosphate. The reaction was cleaned and eluted as before. The reaction was cleaned up using Zymo RCC-5 column and the RNA was eluted in 33.5 µl water. The 5′-diphosphorylated mRNAs were recapped with desthiobiotin-GTP (DTB-GTP, NEB #N0761S) using vaccinia capping enzyme (VCE, NEB #M2080S; 5 µL VCE buffer, 0.5 µL inorganic pyrophosphatase (NEB #M0361S), 5 µL DTB-GTP (5 mM), 50 U VCE, 1 µl SUPERase·In) at 37 °C for 1 hr. The reaction was cleaned up using Zymo RCC-5 column and the RNA was eluted in 30 µl water. Now the RNA is ready for streptavidin enrichment.

5' enrichment. Streptavidin beads (NEB #S1421S) were washed in a high salt wash buffer (10 mM Tris-HCl pH 7.5, 2 M NaCl, 1 mM EDTA) and resuspended in the high-salt buffer at 4 mg/mL. To enrich the RNA and tag the 5' and 3' end by the specific adapter, the 30 µl recapped RNA was first mixed with 30 µl streptavidin beads and incubated at room temperature for 45 min with agitation. Beads were washed twice in high-salt buffer, twice in a lower salt buffer (10 mM Tris-HCl pH 7.5, 250 mM NaCl, 1 mM EDTA), and twice in PNK wash buffer (20 mM Tris-HCl pH 7.5, 10 mM $MgCl_2$, 0.2% Tween). To remove 3' phosphates resulting from fragmentation during conversion, beads were resuspended in 50 µl PNK reaction without ATP (5 µl 10 X PNK buffer, 1 µl T4 PNK (#M0201S), 1 µl SUPERase·In, 43 µl water), and incubate at 37 °C for 30 minutes with agitation. The beads were then washed once in PNK wash buffer, once in 2 M NaCl wash, and twice in PNK wash. Next, RNA was ligated to a 74 nt-long 3' adapter in the following 40 µl 3' ligation mix: 4 µL T4 RNA ligase buffer, 2 µl T4 RNA ligase 2 truncated KQ (NEB #M0373L), 1 µl SUPERase·In, 2 µl extended-L7 adapter (20 µM stock), 16 µl of 50% PEG-8000 and incubated at 25 °C for 2 hr. After incubation, the reaction buffer was removed by washing once with high-salt buffer and twice with PNK wash buffer. To remove the exceeded adapter, the beads were incubated in 50 µl adapter digestion reaction (40 µl water, 5 µl 10 X RNA ligase buffer, 1 µl RecJf [NEB #M0264S], 1 5' Deadenylase [NEB #M0331S], 1 µl SUPERase·In) at 30 °C for 15 min then at 37 °C for 15 min. The beads were washed once with PNK wash, once with high-salt buffer, and twice with low-salt buffer. To elute the DTB-labeled RNA, beads were then suspended with 30 µl low-salt wash buffer containing 1 mM free D-biotin (Thermo Fisher #B20656) and incubate at room temperature for 1 hr. The DTB-labeled RNA was purified with ethanol-AMPure XP (RNA:beads:ethanol = 1:2:3) and eluted in 30 µl water. To increase stringency, the streptavidin bead enrichment was repeated omitting enzymatic steps and instead washing three times with high salt and then three times with lower salt wash, and the eluate was cleaned again by ethanol-AMPure XP.

5' adapter addition. The DTB-GTP cap was removed using 0.5 U/µl RppH (NEB #M0356S) in 1 X ThermoPol buffer (NEB #M0356S) and incubating at 37 °C for 1 hr. The resulting 5'-monophosphate RNA fragments were purified by ethanol-AMPure XP and eluted in 10 µl water. 1 µl (10 pmol) reverse transcription primer was pre-annealed to the templates by heating up to 75 °C for 5 min, then 37 °C for 15 min, 25 °C for 15 min, and chilled at 4 °C. 10 pmol of a 5' adapter was ligated for 3 hr at 25 °C with 2 U/µl T4 RNA ligase 1 (NEB #M0437M). This RNA adapter contains an 8 nt- or 11 nt-long UMI followed by a fixed sequence (AUAU) at its 3' end. The UMI allows robust duplicate removal, and the fixed sequence provides an anchor point to correctly identify the first nucleotide of the mRNA. 40 µl ligation product was used.

cDNA synthesis and PCR. Reverse transcription was carried out at 50 °C for 45 min in a 50 µl reaction containing 0.5 mM dNTPs, 5 mM DTT, 20 U RNaseOUT, 50 mM Tris-HCl pH 8.3, 75 mM KCl, and 300 U SuperScript III. To perform indexing PCR, 40 µl Phusion master mix (NEB # M0532L) was added to the reverse transcription product, along with 5 µl i5 indexing primer and 5 µl i7 indexing primer (NEB #E7600S). Cycling conditions were typically 98 °C 2 min, then 16 cycles of 98 °C 15 s, 65 °C 30 s, 72 °C 30 s, with a final 5 min 72 °C extension. Two rounds of 0.9 X AMPureXP bead purifications were performed to remove primers. Normally ~10 ng indexed library was obtained for each library. The libraries were mixed and sequenced by NovaSeq 6000 or NovaSeqX.

## CROWN-seq data processing

The read pairs were firstly quality trimmed by Cutadapt (*Martin, 2011*): -m 32 -q 20 -e 0.25-a AGAT CGGAAGAGCACACGTC. For the 8 nt-long 5' adapter, -A ATATNNNNNNNNAGATCGGAAGAG CGTCGTG was used; for the 11 nt-long adapter, -A ATATNNNNNNNNNNNAGATCGGAAGAGCGTC GTG was used. Then the UMI along with the fixed ATAT spacer sequences were extracted by UMI-tools (*Smith et al., 2017*). The alignment process was modified from the previous RNA bisulfite alignment strategy (*Huang et al., 2019*). In brief, in silico converted read pairs (read1 A-to-G, read2 T-to-C) were aligned by HISAT2 (*Kim et al., 2019*) against A-to-G converted (for positive strand) and T-to-C converted (for negative strand) reference genome and transcriptome first (key options: `-k 5 -fr -rna-strandness FR -no-temp-splicesite -no-mixed`). Then the unique alignments were extracted and the in silico converted reads were inverse-transformed to the original format. Since two sequences after conversion can be easily confused, we require the best alignment results can be well distinguished from the secondary alignments. Here, the alignment scores (AS tag in Hisat2 alignments, higher is better) of the best alignments should be higher than –10. Meanwhile, the difference

between the best alignments and secondary alignments should be larger than 9. For paired-end alignments, the alignment scores of read1 and read2 were summed. Only read1 was used in the 5' end analysis. Only read1 reads without 5' end softclips were used. Pileup was performed to obtain the read coverages of every 5' end in the transcriptome. Non-conversion rates of the transcription start nucleotides were calculated by A counts over A and G counts.

To annotate the TSNs mapped in CROWN-seq, we used the TSSs in Gencode v45 as the reference TSS positions. We first calculated the distance between the mapped TSNs and the annotated TSSs by BEDtools (*Quinlan and Hall, 2010*). We then tried to assign a TSN to a gene if there was an annotated TSS <100 nt away. Because there can be multiple annotations available, we used the following priority in selecting gene annotations: `snRNA > snoRNA > protein-coding > lncRNA > others`. We also annotated TSNs which come from RNA highly similar to snRNA, snoRNA, or their pseudogenes. To do so, we first built a BLASTn database containing all snRNA, snoRNA, and their pseudogene sequences from Gencode v45. We then performed BLASTn (BLAST 2.9.0+*Camacho et al., 2009*) on the A-TSNs along with the first 50 nt downstream sequences to examine the similarity to the known snRNA, snoRNA, and pseudogenes. The following parameter was used: `-qcov_hsp_perc 50 -perc_identity 50 -word_size 10`. Sequences with bitscore ≥50 were considered as snRNA/snoRNA-like. We also annotated uORF and IRES elements based on uORFdb (*Manske et al., 2023*) and IRES atlas (*Yang et al., 2021*), respectively.

The choice of parameters can significantly affect the accuracy of TSS maps and the precision in $m^6Am$ quantification. In this study, we used several different parameters in defining TSS signals from ReCappable-seq and CROWN-seq. In *Figure 1* and *Figure 1—figure supplement 1*, for the preliminary analyses with ReCappable-seq, we defined TSSs as those with ≥1 TPM coverage as previously used (*Yan et al., 2022*). Notably, this threshold is empirical and subjective for TSS identification. This threshold can result in false negatives, especially for those TSS with expression levels a bit lower than 1 TPM. In *Figure 2* and *Figure 2—figure supplement 1*, to define A-TSNs in CROWN-seq, we first called high-confidence A-TSNs which at least mapped by 20 reads. This threshold was used in a previous $m^5C$ mapping analysis (*Huang et al., 2019*). The ≥20 reads threshold can yield acceptable precision in $m^6Am$ stoichiometry estimation. When an A-TSN is mapped by 20 reads, the quantification precision is 0.05 (1/20). With this threshold, the median coverage of the A-TSNs is ~40–60 among samples, which means precision at 0.017–0.025. Notably, according to the analysis shown in *Figure 2H*, *Figure 2—figure supplement 1I*, CROWN-seq exhibited very high accuracy in TSN mapping even for the TSNs mapped by three reads. Although this threshold allows us to roughly estimate $m^6Am$ stoichiometry, the variability of the quantified stoichiometry can be high when the read depth is low (particularly for A-TSNs with <50 reads). Thus, we used another criterion while generating the $m^6Am$ landscape among different cell lines. For $m^6Am$ landscape profiling (*Figures 3–6* and the corresponding figure supplements), we want to precisely compare the $m^6Am$ stoichiometry between different cell lines. We first merged all the reads from different biological and/or technical replicates to obtain higher read depths for each cell line. We then increased the threshold of sequencing depth so that only A-TSNs mapped by ≥50 reads were quantified. With this threshold, the minimum precision is set to 0.02 (1/50). In practice, this threshold results in medium read coverage at ~130–150 reads, which indicates precision at 0.0067–0.0077. The high coverage also results in low variability in $m^6Am$ quantification between replicates (see *Figure 3—figure supplement 1B*).

The related pipeline and scripts are available at https://github.com/jhfoxliu/CROWN-seq (*Liu, 2025b*).

## RT-qPCR

1 μg total RNA was used as input. The RNA was then mixed with 1 μl Oligo dT(18) (100 pmoles) (Thermo Fisher #SO131), and 1 μl dNTP in 14.5 μl total volume. The mix was incubated at 65 °C for 5 min, then on ice for >30 s. After the incubation, 4 μl 5 X RT mix (Maxima H- buffer, Thermo Fisher #EP0751), 0.5 μl RNaseOUT (Thermo Fisher #10777019), and 1 μl Maxima H- RTase were added to the mix. Reverse transcription was performed at 25 °C for 10 min, then 50 °C for 30 min. After reverse transcription, 1 μl cDNA was used for qPCR. In addition to the cDNA input, the qPCR buffer contains 10 μl Power SYBR Green PCR Master Mix (Thermo Fisher #368577), 0.5 μl forward primer, 0.5 μl reverse primer, and 8 μl water. qPCR was performed based on the standard quantification program in QuantStudio 5 System.

## Gene ontology analysis

Gene ontology analyses were performed with R package ClusterProfiler (*Yu et al., 2012*). p-value cutoffs were set to 0.05 and q-value cutoffs were set to 0.1. 'Cellular Components' and 'Biological Process' terms were analyzed. Importantly, corresponding gene sets, rather than all genes, were used as the backgrounds in term enrichment computation. Since the output terms were normally redundant, terms were de-redundancy by the 'simplify' function in R package GOSemSim (*Yu et al., 2010*; `cutoff = 0.7, by="p.adjust", select_fun = min`).

## Motif analysis

To search for the potential motifs for elements related to transcription initiation, we used 're' package in Python to match specific motifs, which are indicated in the figure legends.

To search for the transcription factor binding sites, we used FIMO (*Grant et al., 2011*) to scan for motifs in HOCOMOCO v11 core motifs database (*Kulakovskiy et al., 2018*).

## RNA secondary prediction and minimal free energy calculation

ViennaRNA package (version 2.5.1) was used to perform RNA secondary structure prediction (*Lorenz et al., 2011*). The RNAfold Python API 'RNA' was used in the analysis. The folding temperature was set to 37 °C. The minimum free energy of the predicted structure was used.

## Quantification and statistical analysis

Quantitative and statistical methods are described above and in figure legends according to their respective technologies and analytic approaches. Statistical analysis and visualization were mainly performed with Python (version 3.8.7). R (version 4.2.2) was used in differential gene expression analysis and Gene Ontology analysis.

Versions of key Python packages: numpy (1.23.5); pandas (1.5.2); scipy (1.9.3); matplotlib (3.6.2); seaborn (0.12.1); matplotlib-venn (0.11.9).

Versions of key R packages: DESeq2 (1.38.1); clusterProfiler (4.6.0); enrichplot (1.18.3); GOSemSim (2.24.0); org.Hs.eg.db (3.16.0).

All boxplots and violin plot summary statistics show the median and IQR of the underlying data. Statistical tests are described in the appropriate figure legends. Student's t-test was applied for two sample non-paired comparisons. One-sided or two-sided testing was performed according to figure legends. If possible, we omitted significance 'stars' from figures; p-values (or equivalent) are instead reported.

## Materials availability

This study did not generate new unique reagents.

## Acknowledgements

We thank members of the Jaffrey lab for their comments and suggestions throughout the duration of this project. We thank members of the Genomics core facility at Weill Cornell Medicine for Illumina sequencing. This work is supported by NIH grants R35 NS111631, S10 OD030335, RM1 HG011563, MH121072, and R01 DA059544 (SRJ).

## Additional information

### Competing interests

Ben R Hawley: Senior scientist in Engage Bio. Samie R Jaffrey: Co-founder and/or has equity in Chimerna Therapeutics, 858 Therapeutics, and Lucerna Technologies. The other authors declare that no competing interests exist.

## Funding

| Funder | Grant reference number | Author |
|---|---|---|
| National Institute of Neurological Disorders and Stroke | R35 NS111631 | Samie R Jaffrey |
| NIH Office of the Director | S10 OD030335 | Samie R Jaffrey |
| National Human Genome Research Institute | RM1 HG011563 | Samie R Jaffrey |
| National Institute of Mental Health | MH121072 | Samie R Jaffrey |
| National Institute on Drug Abuse | R01 DA059544 | Samie R Jaffrey |

The funders had no role in study design, data collection and interpretation, or the decision to submit the work for publication.

### Author contributions

Jianheng Fox Liu, Conceptualization, Resources, Data curation, Software, Formal analysis, Investigation, Visualization, Methodology, Writing – original draft, Writing – review and editing; Ben R Hawley, Resources, Formal analysis; Luke S Nicholson, Resources, Writing – review and editing; Samie R Jaffrey, Conceptualization, Data curation, Supervision, Funding acquisition, Validation, Investigation, Writing – original draft, Project administration, Writing – review and editing

### Author ORCIDs

Jianheng Fox Liu (iD) https://orcid.org/0000-0003-0216-1951
Samie R Jaffrey (iD) https://orcid.org/0000-0003-3615-6958

Reviewer #1 (Public review): https://doi.org/10.7554/eLife.104139.3.sa1
Reviewer #2 (Public review): https://doi.org/10.7554/eLife.104139.3.sa2
Reviewer #3 (Public review): https://doi.org/10.7554/eLife.104139.3.sa3
Author response https://doi.org/10.7554/eLife.104139.3.sa4

# Additional files

### Supplementary files

Supplementary file 1. The overview of CROWN-seq libraries.

MDAR checklist

### Data availability

Sequencing data have been deposited in GEO under accession codes GSE188510 (ReCappable-seq) and GSE233655 (CROWN-seq). Large processed data have been deposited in Zenodo. Source data files have been provided for Figures 2, 3, 5 and 6. All original code has been deposited on GitHub: ReCappable-seq analysis: https://github.com/jhfoxliu/ReCappable-seq (*Liu, 2025a*); GLORI analysis: https://github.com/jhfoxliu/GLORI_pipeline (*Liu, 2023*); CROWN-seq analysis: https://github.com/jhfoxliu/CROWN-seq (*Liu, 2025b*). Any additional information required to reanalyze the data reported in this paper is available from the lead contact upon request.

The following datasets were generated:

| Author(s) | Year | Dataset title | Dataset URL | Database and Identifier |
|---|---|---|---|---|
| Hawley BR, Jaffrey SR | 2024 | mRNA juxtacap sequences govern mRNA translation and stability | https://www.ncbi.nlm.nih.gov/geo/query/acc.cgi?acc=GSE188510 | NCBI Gene Expression Omnibus, GSE188510 |

*Continued*

| Author(s) | Year | Dataset title | Dataset URL | Database and Identifier |
|---|---|---|---|---|
| Liu J, Jaffrey SR, Luke N | 2024 | Absolute transcription-start nucleotide m6Am stoichiometry quantification by CROWN-Seq | https://www.ncbi.nlm.nih.gov/geo/query/acc.cgi?acc=GSE233655 | NCBI Gene Expression Omnibus, GSE233655 |
| Liu J, Jaffrey SR | 2024 | m6Am landscape of human cell lines | https://doi.org/10.5281/zenodo.12760731 | Zenodo, 10.5281/zenodo.12760731 |

The following previously published datasets were used:

| Author(s) | Year | Dataset title | Dataset URL | Database and Identifier |
|---|---|---|---|---|
| Jaffrey SR | 2019 | Identification of the m6Am Methyltransferase PCIF1 Reveals the Location and Functions of m6Am in the Transcriptome | https://www.ncbi.nlm.nih.gov/geo/query/acc.cgi?acc=GSE122948 | NCBI Gene Expression Omnibus, GSE122948 |
| Liu C, Li K | 2022 | Absolute quantification of single-base m6A methylation in the mammalian transcriptome | https://www.ncbi.nlm.nih.gov/geo/query/acc.cgi?acc=GSE210563 | NCBI Gene Expression Omnibus, GSE210563 |
| Goh WS | 2019 | An atlas of single-base-resolution N6-methyl-adenine methylomes redefines RNA demethylase function as suppressors of disruptive RNA methylation | https://www.ncbi.nlm.nih.gov/geo/query/acc.cgi?acc=GSE124509 | NCBI Gene Expression Omnibus, GSE124509 |
| Akichika S, Hirano S, Shichino Y, Suzuki T, Nishimasu H, Ishitani R, Hirose Y, Iwasaki S, Nureki O, Suzuki T | 2018 | Cap-specific terminal N6-methylation of RNA by an RNA polymerase II-associated methyltransferase | https://www.ncbi.nlm.nih.gov/geo/query/acc.cgi?acc=GSE122071 | NCBI Gene Expression Omnibus, GSE122071 |

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
