## [Editor Report · eLife Assessment]

This study presents a new quantitative method, CROWN-seq, to map the cap-adjacent RNA modification N6,2'-O-dimethyladenosine (m6Am) with single nucleotide resolution. Using thoughtful controls and well-validated reagents, the authors provide **compelling** evidence that the method is reliable and reproducible. Additionally, the study provides **important** evidence that m6Am may increase transcription in modified mRNAs. However, the data only demonstrates a correlation between m6Am and transcriptional regulation rather than causality. Overall, this study is poised to advance m6Am research, being of broad interest to the RNA biology and gene regulation fields.

---

## [Referee Report · Reviewer #1 (Public review)]

Summary:

In this manuscript, Liu et al. present CROWN-seq, a technique that simultaneously identifies transcription-start nucleotides and quantifies N6,2'-O-dimethyladenosine (m6Am) stoichiometry. This method is derived from ReCappable-seq and GLORI, a chemical deamination approach that differentiates A and N6-methylated A. Using ReCappable-seq and CROWN-seq, the authors found that genes frequently utilize multiple transcription start sites, and isoforms beginning with an Am are almost always N6-methylated. These findings are consistently observed across nine cell lines. Unlike prior reports that associated m6Am with mRNA stability and expression, the authors suggest here that m6Am may increase transcription when combined with specific promoter sequences and initiation mechanisms. Additionally, they report intriguing insights on m6Am in snRNA and snoRNA and its regulation by FTO. Overall, the manuscript presents a strong body of work that will significantly advance m6Am research.

Strengths:

The technology development part of the work is exceptionally strong, with thoughtful controls and well-supported conclusions.

Weaknesses:

Given the high stoichiometry of m6Am, further association with upstream and downstream sequences (or promoter sequences) does not appear to yield strong signals. As such, transcription initiation regulation by m6Am, suggested by the current work, warrants further investigation.

---

## [Referee Report · Reviewer #2 (Public review)]

Summary:

In the manuscript "Decoding m6Am by simultaneous transcription-start mapping and methylation quantification" Liu and co-workers describe the development and application of CROWN-Seq, a new specialized library preparation and sequencing technique designed to detect the presence of cap-adjacent N6,2'-O-dimethyladenosine (m6Am) with single nucleotide resolution. Such a technique was a key need in the field since prior attempts to get accurate positional or quantitative measurements of m6Am positioning yielded starkly different results and failed to generate a consistent set of targets. As noted in the strengths section below the authors have developed a robust assay that moves the field forward.

Furthermore, their results show that most mRNAs whose transcription start nucleotide (TSN) is an 'A' are in fact m6Am (85%+ for most cell lines). They also show that snRNAs and snoRNAs have a substantially lower prevalence of m6Am TSNs.

Strengths:

Critically, the authors spent substantial time and effort to validate and benchmark the new technique with spike-in standards during development, cross-comparison with prior techniques, and validation of the technique's performance using a genetic PCIF1 knockout. Finally, they assayed nine different cell lines to cross-validate their results. The outcome of their work (a reliable and accurate method to catalog cap-adjacent m6Am) is a particularly notable achievement and is a needed advance for the field.

Weaknesses:

No major concerns were identified by this reviewer.

Mid-level Concerns: All previous concerns were addressed in the revised version

---

## [Referee Report · Reviewer #3 (Public review)]

Summary:

m6Am is an abundant mRNA modification present on the TSN. Unlike the structurally similar and abundant internal mRNA modification m6A, m6Am's function has been controversial. One way to resolve controversies surrounding mRNA modification functions has been to develop new ways to better profile said mRNA modification. Here, Liu et al. developed a new method (based on GLORI-seq for m6A-sequencing), for antibody-independent sequencing of m6Am (CROWN-seq). Using appropriate spike-in controls and knockout cell lines, Liu et al. clearly demonstrated CROWN-seq's precision and quantitative accuracy for profiling transcriptome-wide m6Am. Subsequently, the authors used CROWN-seq to greatly expand the number of known m6Am sites in various cell lines and also determine m6Am stoichiometry to generally be high for most genes. CROWN-seq identified gene promoter motifs that correlate best with high stoichiometry m6Am sites, thereby identifying new determinants of m6Am stoichiometry. CROWN-seq also helped reveal that m6Am does not regulate mRNA stability or translation (as opposed to past reported functions). Rather, m6Am stoichiometry correlates well with transcription levels. Finally, Liu et al. reaffirmed that FTO mainly demethylates m6Am, not of mRNA but of snRNAs and snoRNAs.

Strengths:

This is a well-written manuscript that describes and validates a new m6Am-sequencing method: CROWN-seq as the first m6Am-sequencing method that can both quantify m6Am stoichiometry and profile m6Am at single-base resolution. These advantages facilitated Liu et al. to uncover new potential findings related to m6Am regulation and function. I am confident that CROWN-seq will likely be the gold standard for m6Am-sequencing henceforth.

Weaknesses:

Though the authors have uncovered a potentially new function for m6Am, they need to be clear that without identifying a mechanism, their data might only be demonstrating a correlation between the presence of m6Am and transcriptional regulation rather than causality.

---

## [Author Response]

The following is the authors’ response to the original reviews.

**Public Reviews:**

**Reviewer #1 (Public review):**
Summary:In this manuscript, Liu et al. present CROWN-seq, a technique that simultaneously identifies transcription-start nucleotides and quantifies N6,2'-O-dimethyladenosine (m6Am) stoichiometry. This method is derived from ReCappable-seq and GLORI, a chemical deamination approach that differentiates A and N6-methylated A. Using ReCappable-seq and CROWN-seq, the authors found that genes frequently utilize multiple transcription start sites, and isoforms beginning with an Am are almost always N6-methylated. These findings are consistently observed across nine cell lines. Unlike prior reports that associated m6Am with mRNA stability and expression, the authors suggest here that m6Am may increase transcription when combined with specific promoter sequences and initiation mechanisms. Additionally, they report intriguing insights on m6Am in snRNA and snoRNA and its regulation by FTO. Overall, the manuscript presents a strong body of work that will significantly advance m6Am research.Strengths:The technology development part of the work is exceptionally strong, with thoughtful controls and well-supported conclusions.

We appreciate the reviewer for the very positive assessment of the study. We have addressed the concerns below.

Weaknesses:Given the high stoichiometry of m6Am, further association with upstream and downstream sequences (or promoter sequences) does not appear to yield strong signals. As such, transcription initiation regulation by m6Am, suggested by the current work, warrants further investigation.

We thank the reviewer for the insightful comments. We have softened the language related to m^6^Am and transcription regulation. We totally agree with the reviewer that future investigation is required to determine the molecular mechanism behind m^6^Am and transcription regulation.

**Reviewer #2 (Public review):**
Summary:In the manuscript "Decoding m6Am by simultaneous transcription-start mapping and methylation quantification" Liu and co-workers describe the development and application of CROWN-Seq, a new specialized library preparation and sequencing technique designed to detect the presence of cap-adjacent N6,2'-O-dimethyladenosine (m6Am) with single nucleotide resolution. Such a technique was a key need in the field since prior attempts to get accurate positional or quantitative measurements of m6Am positioning yielded starkly different results and failed to generate a consistent set of targets. As noted in the strengths section below the authors have developed a robust assay that moves the field forward.Furthermore, their results show that most mRNAs whose transcription start nucleotide (TSN) is an 'A' are in fact m6Am (85%+ for most cell lines). They also show that snRNAs and snoRNAs have a substantially lower prevalence of m6Am TSNs.Strengths:Critically, the authors spent substantial time and effort to validate and benchmark the new technique with spike-in standards during development, cross-comparison with prior techniques, and validation of the technique's performance using a genetic PCIF1 knockout. Finally, they assayed nine different cell lines to cross-validate their results. The outcome of their work (a reliable and accurate method to catalog cap-adjacent m6Am) is a particularly notable achievement and is a needed advance for the field.Weaknesses:No major concerns were identified by this reviewer.

We thank the reviewer for the positive assessment of the method and dataset. We have addressed the concerns below.

Mid-level Concerns:(1) In Lines 625 and 626, the authors state that “our data suggest that mRNAs initate (mis-spelled by authors) with either Gm, Cm, Um, or m6Am.” This reviewer took those words to mean that for A-initiated mRNAs, m6Am was the ‘default’ TSN. This contradicts their later premise that promoter sequences play a role in whether m6Am is deposited.

We thank the reviewer for the comment. We have changed this sentence into “Instead, our data suggest that mRNAs initiate with either Gm, Cm, Um, or Am, where Am are mostly m^6^Am modified.” The revised sentence separates the processes of transcription initiation and m^6^Am deposition, which will not confuse the reader.

(2) Further, the following paragraph (lines 633-641) uses fairly definitive language that is unsupported by their data. For example in lines 637 and 638 they state “We found that these differences are often due to the specific TSS motif.” Simply, using ‘due to’ implies a causative relationship between the promoter sequences and m6Am has been demonstrated. The authors do not show causation, rather they demonstrate a correlation between the promoter sequences and an m6Am TSN. Finally, despite claiming a causal relationship, the authors do not put forth any conceptual framework or possible mechanism to explain the link between the promoter sequences and transcripts initiating with an m6Am.(3) The authors need to soften the language concerning these data and their interpretation to reflect the correlative nature of the data presented to link m6Am and transcription initiation.

For (2) and (3). We have softened the language in the revised manuscript. Specifically, for lines 633-641 in the original manuscript, we have changed “are often due to” into “are often related to” in the revised manuscript, which claims a correlation rather than a causation.

**Reviewer #3 (Public review):**
Summary:m6Am is an abundant mRNA modification present on the TSN. Unlike the structurally similar and abundant internal mRNA modification m6A, m6Am’s function has been controversial. One way to resolve controversies surrounding mRNA modification functions has been to develop new ways to better profile said mRNA modification. Here, Liu et al. developed a new method (based on GLORI-seq for m6A-sequencing), for antibody-independent sequencing of m6Am (CROWN-seq). Using appropriate spike-in controls and knockout cell lines, Liu et al. clearly demonstrated CROWN-seq’s precision and quantitative accuracy for profiling transcriptome-wide m6Am. Subsequently, the authors used CROWN-seq to greatly expand the number of known m6Am sites in various cell lines and also determine m6Am stoichiometry to generally be high for most genes. CROWN-seq identified gene promoter motifs that correlate best with high stoichiometry m6Am sites, thereby identifying new determinants of m6Am stoichiometry. CROWN-seq also helped reveal that m6Am does not regulate mRNA stability or translation (as opposed to past reported functions). Rather, m6Am stoichiometry correlates well with transcription levels. Finally, Liu et al. reaffirmed that FTO mainly demethylates m6Am, not of mRNA but of snRNAs and snoRNAs.Strengths:This is a well-written manuscript that describes and validates a new m6Am-sequencing method: CROWN-seq as the first m6Am-sequencing method that can both quantify m6Am stoichiometry and profile m6Am at single-base resolution. These advantages facilitated Liu et al. to uncover new potential findings related to m6Am regulation and function. I am confident that CROWN-seq will likely be the gold standard for m6Am-sequencing henceforth.Weaknesses:Though the authors have uncovered a potentially new function for m6Am, they need to be clear that without identifying a mechanism, their data might only be demonstrating a correlation between the presence of m6Am and transcriptional regulation rather than causality.

We thank the reviewer for the very positive assessment of the CROWN-seq method. We have softened the language which is related to the correlation between m^6^Am and transcription regulation.

**Reviewer recommendations:**

We thank the reviewers for their constructive suggestions. In the revised manuscript, we have corrected the errors and updated the requested discussions and figures.

**Reviewer #1 (Recommendations for the authors):**
(1) The prior work from the research group, "Reversible methylation of m6Am in the 5′ cap controls mRNA stability" (PMID: 28002401), should be cited, even if the current findings differ from earlier conclusions-particularly in line 58 and the section titled "m6Am does not substantially influence mRNA stability or translation".

We thank the reviewer for this comment. We have added the citation.

(2) I wonder why the authors chose to convert A to I before capping and recapping, as RNA fragmentation caused by chemical treatment may introduce noise into these processes.

We thank the reviewer for this comment. This is a very good point. We have indeed considered this alternative protocol. There are two concerns in performing decapping-and-recapping before A-to-I conversion: (1) it is unclear whether the 3’-desthiobiotin, which is essential for the 5’ end enrichment, is stable or not during the harsh A-to-I conversion; (2) performing decapping-and-recapping first requires more enzyme and 3’-desthiobiotin-GTP, which are the major cost of the library preparation. This is because the input of CROWN-seq (~1 μg mRNA) is much higher than that in ReCappable-seq (~5 μg total RNA or ~250 ng mRNA). In the current protocol, many 5’ ends are highly fragmented and therefore are lost during the A-to-I conversion. As a result, less enzyme and 3’-desthiobiotin-GTP are needed.

(3) During CROWN-seq benchmarking, the authors found that 93% of reads mapped to transcription start sites, implying a 7% noise level with a spike-in probe. This noise could lead to false positives in TSN assignments in real samples. It appears that additional filters (e.g., a known TSS within 100 nt) were applied to mitigate false positives. If so, I recommend that the authors clarify these filters in the main text.

We thank the reviewer for this comment. We think that the spike-in probes might lead to an underestimation of the accuracy of TSN mapping. The spike-in probes are made by in vitro transcription with m^7^Gpppm^6^AmG or m^7^GpppAmG analogs. We found that the in vitro transcription exhibits a small amount of non-specific initiation, which leads to spike-in probes with 5’ ends that are not precisely aligned with the desired TSS. To better illustrate the mapping accuracy of CROWN-seq, we provided Figure 2H, which compares the non-conversion rates of newly found A-TSNs between wild-type and *PCIF1* knock cells. If the newly found A-TSNs are real, they should show high non-conversion rates in wild-type cells (i.e., high m^6^Am) and almost zero non-conversion rates (i.e., Am) in *PCIF1* knockout cells. As expected, most of the newly found A-TSNs are true A-TSNs since they are m6Am in wild-type and Am in *PCIF1* knockout. Thus, we think that CROWN-seq is very precise in TSS mapping. We have clarified this in the Discussion.

(4) I wonder if PCIF1 knockout affects TSN choice and abundance. If not, this data should be presented. If so, how are these changes accounted for in Figure 2H and Figure S5?

We thank the reviewer for this comment. PCIF1 KO does not really affect TSN choice. Here we calculate the correlation of relative TSN expression within genes between wild-type and PCIF1 KO cells (shown using Pearson’s r). It shows that most of the genes have similar TSN choices (with higher Pearson’s r) in both wild-type and PCIF1 KO cells. Thus, PCIF1 KO does not alter global TSN expressions.

**Author response image 1. sa4fig1:** 

(5) The manuscript refers to Am as a rare modification in mRNA (e.g., introduction lines 101-102; discussion lines 574, 608; and possibly other locations) without specifying this only applies to transcription start sites. As this study does not cover entire mRNA sequences, these statements may not be misleading.

We thank the reviewer for this comment. We have clarified it.

**Reviewer #2 (Recommendations for the authors):**
(1) On line 122, the authors state that: "On average, a gene uses 9.5{plus minus}9 (mean and s.d., hereafter) TSNs (Figure 1A)." However, they do not discuss the dispersion apparent in the TSNs they observed. Figure panels 1A, B, and S1A, B show a range of 120 bases or less. What is the predominant range of distances between annotated TSNs and the newly identified ones?(1a) For example, what percentage of new TSNs fall within 20? 50? 75? bases of the annotated sites? Additional text describing the distribution of these TSNs would help readers better understand the diversity inherent in these novel 5' RNA ends. Notably, this additional text likely is best placed in the CROWN-Seq section related to Figure 2 or S2.

We thank the reviewer for this comment. We have updated Figure S2 to describe the newly found TSSs. Depending on the coverage in CROWN-seq, the TSSs with higher coverage tend to overlap with or locate proximally to known TSSs. In contrast, the TSSs with low coverage tend to be located further away from annotated TSSs.

(1b) The alternate TSNs can have effects on splicing patterns and isoform identity. Providing a few sentences to explain how regularly this occurs would be helpful.

We thank the reviewer for this comment. It is a very interesting point. Different TSNs can indeed have different splicing patterns. Although the discovery of splicing patterns regulated by TSNs is out of the scope of this study, we have discussed this possibility in the revised Discussion section.

(2) On Lines 241 and 242, the authors mentioned that 1284 sites were excluded from the analysis based on low (under 20-explained in the figure legend) read count, distance from TSS, or false negatives (which are not explained). Although I agree that the authors are justified in setting these reads aside, the information could be useful to readers willing to perform follow-up work if their mRNAs of interest were included in these 1284 sites.(2a) An annotation of all of these sites (broken down by category, i.e. the 811, the 343, and the 130) as a supplementary table should be provided.

We thank the reviewer for this comment. We have added the categories to the revised Table S1.

(3) Although I have marked several typos/grammar mistakes in several parts of this review, others exist elsewhere in the text and should be corrected.

We thank the reviewer for this comment. We have corrected them.

(4) In lines 122 and 123 the authors say "Only ~9% of genes contain a single TSN (Figure 1A)." However, their figure shows 81% with a single TSN. Why is there a 10% discrepancy?

We thank the reviewer for this comment. We have corrected the plot in Figure 1A, to match the description.

(5) The first Tab of Table S2 is labeled 'Legend', but is blank. Is this intentional?

We thank the reviewer for this comment. We have updated the table legends.

(6) On lines 70 and 76 of the supplementary figure file pertaining to Figure S2, the legend labels for Figure S2E and S2F are not accurate, they need to be changed to G and H.(7) In Figure 4A 'percentile' is misspelled.(8) The color-coding legend for the 4 bases is missing from (and should be added to) Figure S4A.(9) On Lines 984, 1163, and 1194 the '2s' should be properly sub-scripted where appropriate.

For (6) to (9). We thank the reviewer for finding these issues. We have now corrected them.

**Reviewer #3 (Recommendations for the authors):**
(1) The authors should discuss if their results can definitively distinguish between the SSCA+1GC motif promoting m6Am that, in turn, promotes transcription, versus the SCA+1GC motif promoting m6Am but also separately promoting transcription in a m6Am-independent manner. The authors should also discuss this in light of recent findings by An et al. (2024 Mol. Cell), which support the former conclusion.

We thank the reviewer for the suggestion. We now have updated the Discussion to address that our paper and An et al. can support each other.

(2) Given that the authors showed m6Am promotes gene expression (Figure 5) but does not affect mRNA stability (Fig. S5), logic dictates that m6Am must regulate mRNA transcription. However, the authors should explain why this regulation focuses on the initiation aspect of transcription rather than other aspects of transcriptional e.g. premature termination, pause release, and elongation.

We thank the reviewer for this comment. In this study, we did not profile the 3’ ends of nascent RNAs and thus we can only make conclusions about the overall transcription process but not a specific aspect. We have updated the revised Discussion section to mention that An et al. discovered that m^6^Am can sequester PCF11 and thus promote transcription, and therefore some of the effects we see could be related to differential premature termination.

(3) Authors should add alternative versions of Figure 1D but with 3 colours corresponding to Am vs. m6Am vs. Cm/Gm/Um for all the cells, they performed CROWN-seq on.

We thank the reviewer for this comment. We have updated Figure S5 as the corresponding figure showing the fraction of Am vs. m6Am vs. Cm/Gm/Um.

(4) Figure 2H (left): Please comment on the few outliers that still show high non-conversion even in PCIF1-KO cells.

We thank the reviewer for this comment. We have discussed the outliers in the main text. These outliers can be found in the revised Table S3.

(5) Line 254: "Second, if these sites were RNA fragments they would not contain m6Am." is missing a comma.(6) S2G and S2H labelling in Figure S2 legends is wrong.

For (5) and (6). We thank the reviewer for these comments. We have corrected them.

(7) Figure 3D: Many gene names are printed multiple times (e.g. ACTB is printed 5 times). Is this correct; is each dot representing 1 cell line?

We thank the reviewer for this comment. These gene names represent different transcription-start nucleotides. We now clarify that each instance refers to a different start site.

(8) S5A-C: Even if there's no substantial difference, authors should still display the Student's T-test P-values as they did for S5D-G.

We thank the reviewer for this comment. We have updated the P-values.

(9) Figure 5C and S5E: Why are the authors not showing the respective analysis for C-TSN and U-TSN genes?

We thank the reviewer for this comment. Most mRNAs start with A or G. We therefore selected G-TSN as the control. Unlike G-TSNs which occur in diverse sequence and promoter contexts, C-TSNs and U-TSNs are unusual. Genes that mainly use C-TSNs and U-TSNs are the so-called “5’ TOP (Terminal OligoPyrimidine)” genes. The 5’ TOP genes are mostly genes related to translation and metabolism, and thus their expressions reflect the homeostasis of cell metabolism. Thus, we were concerned that any differential expression of the C-TSN and U-TSN genes between wild-type and PCIF1 knockout cells might reflect specific effects on TOP transcriptional regulation rather than the general effects of PCIF1 on transcription.

(10) Line 82, 470, 506, 676: The authors should also cite Koh et al (2019 Nat. Comm.) in these lines that describe how snRNAs can also be m6Am-methylated and how FTO targets these same snRNAs for demethylation.

We thank the reviewer for this comment. We have updated the citation.